# UNIFIED BRAIN SURFACE AND VOLUME REGISTRATION

**S. Mazdak Abulnaga**[1,2]  **Andrew Hoopes**[1,2]  **Malte Hoffmann**[2]  **Robin Magnet**[3]
**Maks Ovsjanikov**[4]  **Lilla Zöllei**[2]  **John Guttag**[1]  **Bruce Fischl**[2*]  **Adrian V. Dalca**[1,2*]
[1]MIT Computer Science and Artificial Intelligence Laboratory  [2]MGH, Harvard Medical School
[3] Université Paris Cité, INRIA   [4]LIX, CNRS, École Polytechnique
abulnaga@csail.mit.edu

## ABSTRACT

Accurate registration of brain MRI scans is fundamental for cross-subject analysis in neuroscientific studies. This involves aligning both the cortical surface of the brain and the interior volume. Traditional methods treat volumetric and surface-based registration separately, which often leads to inconsistencies that limit downstream analyses. We propose a deep learning framework, NeurAlign, that registers 3D brain MRI images by jointly aligning both cortical and subcortical regions through a unified volume-and-surface-based representation. Our approach leverages an intermediate spherical coordinate space to bridge anatomical surface topology with volumetric anatomy, enabling consistent and anatomically accurate alignment. By integrating spherical registration into the learning, our method ensures geometric coherence between volume and surface domains. In a series of experiments on both in-domain and out-of-domain datasets, our method consistently outperforms both classical and machine learning-based registration methods—improving the Dice score by up to 7 points while maintaining regular deformation fields. Additionally, it is orders of magnitude faster than the standard method for this task, and is simpler to use because it requires no additional inputs beyond an MRI scan. With its superior accuracy, fast inference, and ease of use, NeurAlign sets a new standard for joint cortical and subcortical registration.

## 1 INTRODUCTION

Accurate alignment of both cortical *and* subcortical regions is essential for comprehensive and consistent whole-brain analysis in neuroimaging studies. For example, functional MRI (fMRI) measures brain activity by detecting blood oxygenation changes, with a key goal of localizing functional regions in the cerebral cortex. Mapping activation patterns to brain structure reveals functional organization and is essential for quantifying how neurodegenerative diseases affect things such as cognition (Fischl et al., 2008; Thompson et al., 2007; Ghosh et al., 2010).

Traditional deformable volumetric registration methods compute a three-dimensional (3D) displacement field to align brain images by maximizing intensity similarity (Avants et al., 2008; Balakrishnan et al., 2019; Rueckert et al., 1999). While effective for aligning subcortical structures and global anatomy, volumetric deformable registration often fails in the cortex. The cortex is a thin, highly curved surface with significant inter-subject variability in folding patterns that is difficult to align in Euclidean space (Fischl et al., 1999). Accurate alignment is critical, since the cortex encodes many of the brain's high-level functions such as memory and language, and serves as a primary target for structural and functional neuroimaging studies (Fischl, 2012). Surface-based methods address this challenge by projecting the cortex onto a sphere and aligning the cortex using pre-computed anatomical features such as sulcal depth and/or curvature. This facilitates registration by providing a common geometric space to preserve the topology of the cortex (Fischl et al., 1999; Yeo et al., 2009).

Using fundamentally different representations for surface and volume registration forces neuroscience researchers to tackle two disjoint problems and combine solutions *ad hoc* (Tucholka et al., 2012). This

---

*co-senior authors

typically involves solving an elastic partial differential equation to propagate the surface registration to the interior brain. However, interpolating the surface registration to the interior is not guaranteed to be consistent with the original volumetric registration. This sequential approach prevents computing a coherent registration that simultaneously satisfies both cortical and subcortical alignment objectives. This introduces errors, undermining whole-brain analyses by potentially misaligning anatomically and functionally connected areas (Joshi et al., 2009; Postelnicu et al., 2008; Ahmad et al., 2019).

In this work, we propose `NeurAlign`, a machine learning framework for unified registration of cortical and subcortical structures. NeurAlign, simultaneously trains a volumetric registration network and a spherical registration network using a shared framework that combines both domains. Unlike existing approaches, our model explicitly couples the two through a loss function that encourages the volumetric deformation field on the cortical ribbon to match the spherical registration field. This promotes topologically correct and geometrically consistent alignment across cortical and subcortical regions. Crucially, because our method performs registration in a single forward pass, it enables efficient and scalable whole-brain registration for large population studies. Further, NeurAlign does not require cortical meshes nor segmentations at inference, avoiding the computationally intensive preprocessing required by other methods. Our method, NeurAlign, achieves significantly improved cortical and subcortical alignment compared to the standard joint registration framework, CVS (Postelnicu et al., 2008), while reducing computation time by several orders of magnitude. In addition, NeurAlign substantially outperforms state-of-the-art volumetric machine learning–based registration approaches. To summarize our contributions:

- We propose a unified machine learning model that performs consistent registration of both cortical and subcortical structures using coupled spherical and volumetric networks.
- We derive a cortical consistency loss that explicitly encourages agreement between the volumetric deformation field in the cortex and the spherical registration field, promoting anatomically coherent alignment across the entire brain.
- Our proposed method leverages cortical meshes during *training*, but does not require them during inference, enabling accessible, rapid, and accurate registration.
- We validate our method on four clinical neuroimaging datasets, showing substantially improved and rapid cortical and subcortical alignment compared to standard baselines.

Our code and model weights are available at `https://github.com/mabulnaga/neuralign`.

## 2 RELATED WORK

**Volumetric Registration.** Deformable registration estimates a dense spatial mapping between a pair of images. Classical methods (Rueckert et al., 1999; Ashburner, 2007; Rohr et al., 2001; Modat et al., 2010) solve an optimization problem for each pair of images. This balances an image-similarity objective, often mean-squared error (MSE) or normalized cross correlation (NCC) (Avants et al., 2008), between the warped image and the fixed image, with a regularization objective that encourages a smooth solution. Methods differ in their optimization strategy, choice of the objective function, and in how they parameterize the deformation field.

Learning-based methods fit the parameters of a neural network to a training dataset. The network learns a nonlinear function that maps an input image pair to an output transform, generally enabling much faster inference than classical methods. Supervised approaches (Sokooti et al., 2017; Rohé et al., 2017; Yang et al., 2017) reproduce known simulated deformations or fields estimated by classical methods, whereas unsupervised approaches (De Vos et al., 2017; Balakrishnan et al., 2019; Mok et al., 2023; Grzech et al., 2022; Abulnaga et al., 2025; Hoffmann et al., 2021; Rakic et al., 2026) optimize an image-similarity loss term and a regularization loss term at training, analogous to classical registration.

Both optimization-based and machine learning-based volumetric approaches are highly effective at aligning subcortical structures. However, they struggle with cortical registration, since the optimization landscape in Euclidean space often leads to suboptimal local minima. This can lead to implausible solutions that, for example, move voxels from one gyrus to the next instead of following the cortical fold. Further, relying only on image-based features cannot align the functional regions

of the cortex, so geometric features must be employed from geometric representations (Fischl et al., 1999; Ségonne et al., 2007).

**Spherical Registration.**    Cortical alignment in the spherical domain is the standard approach in neuroscientific studies (Fischl, 2012; Ghosh et al., 2010; Tucholka et al., 2012). Mapping the cortex to the sphere simplifies the optimization, since the registration is guaranteed to preserve topology. The registration is driven by geometric descriptors that measure the cortical shape and folding patterns (Fischl et al., 1999; Yeo et al., 2009; Conroy et al., 2013; Zhao et al., 2019). Similar to volumetric registration, spherical registration is posed as an optimization problem that balances a geometric similarity objective with a regularization objective. Recent methods use deep learning to improve inference time (Li et al., 2024; Zhao et al., 2019; Cheng et al., 2020). The spherical mesh is parameterized to the 2D plane using a stereographic projection. The registration is performed in 2D space using convolutional neural networks (Zhao et al., 2019). Icosahedral CNNs (Zhao et al., 2021) learn directly on the sphere but require a rigid tessellation that is not produced natively by standard neuroimaging pipelines. These methods have achieved comparable accuracy to classical techniques with a significant improvement in speed. We build on the planar parameterization methods to construct a spherical registration network and jointly model it with a volumetric network.

**Surface Mesh Registration.**    Surface registration aligns two meshes or point clouds in 3D Euclidean space. Classical approaches typically frame the task as a deformation problem, aiming to align two shapes by estimating a rigid or non-rigid transformation, that minimizes a distance metric (Besl & McKay, 1992; Rusinkiewicz & Levoy, 2001; Sorkine & Alexa, 2007; Weischedel, 2012; Abulnaga et al., 2023). An alternative paradigm treats surface registration as a *shape correspondence* problem, focusing on establishing point-to-point correspondences based on intrinsic geometry (Deng et al., 2022). Functional maps (Ovsjanikov et al., 2012) and extensions (Sharp et al., 2022; Donati et al., 2020; Cao et al., 2023) have emerged as a powerful representation for such correspondences, enabling compact linear descriptions of dense mappings between shapes. However these methods, developed for graphics applications target moderate complexity meshes and scale poorly to high-curvature, high-resolution cortical surfaces, despite some efforts for memory-efficient variants (Magnet & Ovsjanikov, 2024). Furthermore, because of the high curvature and topological consistency constraints of cortical surfaces, Euclidean 3D embeddings become unstable and less informative. Finally, these methods are unsuitable for registering volumetric structures as they are only defined for 2-dimensional surfaces. These reasons further motivate the use of spherical registration to align cortical surfaces.

**Combined Volume and Surface Registration.**    Existing neuroscientific studies treat volumetric and cortical registration as two separate problems. The Combined Volume and Surface framework (CVS) (Postelnicu et al., 2008; Zöllei et al., 2010) is the standard method for jointly aligning cortical and subcortical anatomy in adult brains. CVS first aligns the cortical surface then propagates the resulting deformation into the volume using an elastic model, followed by intensity-based refinement. While effective, this sequential formulation decouples the surface and volumetric objectives, can introduce inconsistencies near the cortical–subcortical boundary, and requires surface meshes and long runtimes. Related work has also used cortical surface registration to supervise volumetric registration (Ahmad et al., 2019), though focused in the infant-brain domain. A strength of these variational approaches is that they offer a principled way to propagate surface constraints into the volume using well-understood physical models. However, they remain impractical for large-scale datasets, as they require several hours of computation per pair and depend on extracting cortical surfaces and segmentation labelmaps.

To address these limitations, we introduce NeurAlign, which incorporates the established idea of surface-guided alignment into a single, unified learning-based diffeomorphic framework. NeurAlign jointly aligns cortical and subcortical structures through a single forward pass, requires only 3D MRI at inference time (without surface meshes), and provides the speed and simplicity needed for high-throughput neuroimaging studies.

## 3    METHODS

We develop NeurAlign, a method for jointly aligning cortical and subcortical structures in 3D brain MRI. We jointly optimize a volumetric and a spherical deformation field and propose a soft constraint

to encourage consistency between the two registration fields. We first propose the continuous formulation of the problem before describing the discrete case.

## 3.1 CONTINUOUS MODEL

Let $M_1, M_2 \subset \mathbb{R}^3$ be bounded brain volumes, with associated smooth cortical surfaces $\partial M_1, \partial M_2$. We seek a bijective map $\varphi : M_1 \to M_2$ that aligns both the cortical and subcortical structures. Each volume is equipped with an intensity sampling function $I_i : \mathbb{R}^3 \to \mathbb{R}$. The surfaces are also equipped with $d$ geometric descriptor functions $g_i : \partial M_i \to \mathbb{R}^d$, such as sulcal depth or mean curvature.

Finding $\varphi$ amounts to finding a solution to the following optimization problem,

$$\underset{\varphi \in \mathcal{P}}{\arg \min} \int_{M_1} f_I \left( I_1(\mathbf{x}), I_2\big(\varphi(\mathbf{x})\big) \right) dV(\mathbf{x}) + \int_{\partial M_1} f_S \left( g_1(\mathbf{x}), g_2\big(\varphi(\mathbf{x})\big) \right) dS(\mathbf{x}) + \mathrm{R}[\varphi], \quad (1)$$

with $\mathcal{P}$ denoting the feasible set of maps, enforcing for instance $\varphi(\partial M_1) \subset \partial M_2$.

Here, $f_I$ measures image similarity, $f_S$ compares surface descriptors, $dV$ and $dS$ denote the volume and surface measures, and $\mathrm{R}[\cdot]$ denotes a set of regularization functionals.

Solving Eq. (1) is difficult: the joint matching of intensities and geometric features has a nonconvex objective with degenerate solutions. Enforcing $\varphi(\partial M_1) \subseteq \partial M_2$ is especially challenging as the cortex is thin, highly folded, and has variable structure across subjects. Fine-grained boundary correspondence is poorly captured in a Euclidean volumetric grid, and optimizers often only align subsets of folds while blurring or misaligning others (Ségonne et al., 2007). Finally, finding a unified representation that captures both the volumetric image and cortical surface is difficult.

We use a joint representation, where cortical alignment is performed on the sphere, using topology-preserving maps and volumetric registration is performed in 3D. By enforcing consistency between the two, this learning-based formulation efficiently registers both the cortical and subcortical structures.

## 3.2 SPHERICAL ALIGNMENT

We revise the optimization problem in Eq. (1) by introducing a spherical intermediate domain to help enforce the constraint set $\mathcal{P}$. Let $\tau_i : \partial M_i \to \mathbb{S}^2$ denote the (fixed) invertible spherical maps obtained via cortical inflation (Fischl et al., 1999; Hoopes et al., 2022) for each surface. The cortical alignment problem then computes a map $\psi : \mathbb{S}^2 \to \mathbb{S}^2$ that displaces spherical coordinates $(\theta, \phi)$, to align geometric descriptors. By optimizing on the sphere, we guarantee satisfaction of the constraint $\mathcal{P}$. The revised optimization problem is then:

$$\underset{\varphi, \psi \in \mathcal{Q}}{\arg \min} \int_{M_1} f_I \left( I_1(\mathbf{x}), I_2\big(\varphi(\mathbf{x})\big) \right) dV(\mathbf{x}) + \int_{\partial M_1} f_S \left( g_1\big(\tau(\mathbf{x})\big), g_2\big(\psi(\tau(\mathbf{x}))\big) \right) dS(\tau(\mathbf{x})) + \mathrm{R}[\varphi] + \mathrm{R}[\psi],$$
$$(2)$$

We decouple this optimization into two separate optimization problems: volumetric registration of subcortical regions using $\varphi$, and cortical registration on the sphere using $\psi$. We introduce a constraint set $\mathcal{Q}$ to encourage a consistent mapping below.

**Cortical Alignment Consistency Constraint.** Optimizing Eq. (2) independently can estimate maps $\varphi$ and $\psi$ that map the cortex to different locations in the volume. We therefore require consistency in Euclidean space by composing maps to the sphere and back: $\mathcal{Q} = \{\varphi(\mathbf{x}) = \tau_2^{-1} \circ \psi \circ \tau_1(\mathbf{x})$ for $\mathbf{x} \in \partial M_1\}$.

To keep the problem tractable, we soften this constraint using a coupling energy that encourages consistent alignment:

$$E_{cons}(\varphi, \psi) = \int_{\partial M_1} f \left( \varphi(\mathbf{x}), \tau_2^{-1}(\psi(\tau_1(\mathbf{x}))) \right) dS(\mathbf{x}), \quad (3)$$

where $f$ measures dissimilarity in the mapping by the squared error. We replace the hard constraint $\mathcal{Q}$ with the coupling energy (Eq. (3)) to achieve a consistent mapping that accurately aligns both cortical and subcortical structures. This constraint only acts on a set of Lebesgue measure zero in $\mathbb{R}^3$

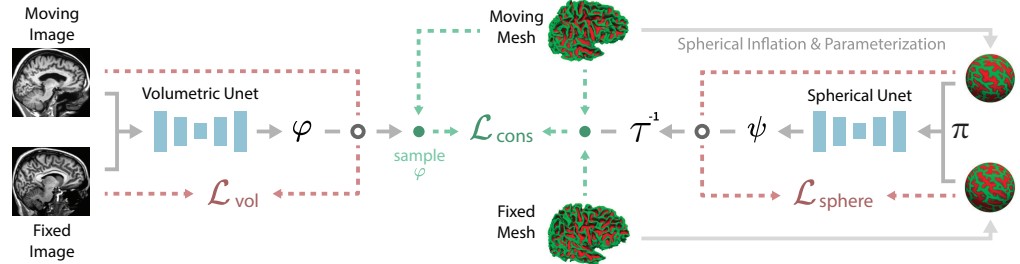

Figure 1: Unified cortical and volumetric registration framework. We employ a volumetric CNN to perform 3D image alignment, and a 2D spherical CNN to align the cortical surfaces in the spherical domain. In training, the models require the MRI images, the cortical surface meshes, and the inflated spheres. The resultant deformation fields of both models are used in a loss term ($\mathcal{L}_{cons}$) that encourages cortical registration to be consistent in the spherical and volumetric pathways. At inference time, the 3D model does not require meshes nor spheres to perform registration.

(the cortical surface $\partial M_1$) and is therefore weak. In practice, however, our discrete implementation avoids this issue: the consistency loss samples the volumetric deformation at mesh vertex locations via trilinear interpolation, distributing gradients to neighboring voxels. We also impose smoothness priors on the map, allowing the surface-derived updates to naturally propagate throughout the volume.

The remainder of this section focuses on the discretization and use of neural networks to model the registration problem.

## 3.3 DISCRETE MODEL

We discretize the optimization problem in Eq. (2), and use neural networks to parameterize the volumetric and spherical maps. Specifically, we formulate an unsupervised learning problem where we jointly learn 3D and 2D U-Net convolutional neural networks (CNN) (Ronneberger et al., 2015). Given an image and a surface, the networks output the 3D and 2D deformation fields $\varphi, \psi$, respectively. We propose a consistency loss to obtain an accurate cortical surface registration in the volume. Figure 1 presents an overview of our framework.

**Notation.** Let $I_1, I_2 : \mathbb{R}^3 \to \mathbb{R}$ denote two 3D brain MRI scans, where $I_i(\mathbf{x})$ gives the intensity at spatial location $\mathbf{x}$. Each image used in training the networks also has an associated triangular surface mesh $\mathcal{C}_1, \mathcal{C}_2 \subset \mathbb{R}^3$ representing the cortical surface extracted from the respective anatomy. Vertex $j$ of mesh $i$, $\boldsymbol{v}_j^i$, is a 3D coordinate in the image space of $I_i$. Each cortical surface mesh has an associated mesh in the spherical domain denoted $\mathcal{S}_1, \mathcal{S}_2$, where $\mathcal{S}_1 = \tau_1(\mathcal{C}_1)$, with the same triangle connectivity. The vertices $\boldsymbol{v}$ are mapped to the sphere by a standard spherical inflation procedure and have an associated $\mathbb{S}^2$ representation (Fischl et al., 1999).

**Volumetric Registration Network.** We employ a neural network $F_v(I_1, I_2; \omega_v) = \varphi$ to predict $\varphi$ as a function of the input images, where $F_v$ is parameterized by $\omega_v$. A core 3D convolutional network first outputs a stationary velocity field (SVF), which we then integrate to obtain a diffeomorphic displacement field $\varphi$ (Ashburner, 2007; Dalca et al., 2019a). In the deformation field $\varphi : \mathbb{R}^3 \to \mathbb{R}^3$, each voxel in $\mathbb{R}^3$ is assigned a displacement vector $\boldsymbol{u}$: $\varphi(\boldsymbol{x}) = \boldsymbol{x} + \boldsymbol{u}(\boldsymbol{x})$.

**Spherical Registration Network.** We construct a 2D CNN registration network for the sphere. We parameterize the spherical meshes to the 2D Euclidean plane using a stereographic projection $\pi : \mathbb{S}^2 \to \mathbb{R}^2$. By mapping to the plane, we can use a standard 2D CNN to learn a 2D diffeomorphic displacement field in the space of spherical angles $(\theta, \phi)$.

To account for the nonuniform sampling of the spherical parameterization, where regions near the poles are stretched, we apply distortion correction in all surface-based losses by weighting samples by $\sin(\theta)$, where $\theta$ is the elevation angle (Cheng et al., 2020; Li et al., 2024). We also handle the discontinuities at the image boundaries using the padding strategy of (Li et al., 2024), employing circular padding along the left–right axis and a $180°$ circular shift with reflection padding at the poles.

We use a neural network $F_s(\pi(\mathcal{S}_1), \pi(\mathcal{S}_2); \omega_s) = \psi$ to predict $\psi$, by first predicting a SVF that is integrated. The network outputs a displacement field $\boldsymbol{u}_s$ such that $\psi(\rho) = \rho + \boldsymbol{u}_s(\rho)$, where $\rho = (\theta, \phi)$ is a discrete location in pixel coordinates on the parameterized grid.

### 3.3.1 Unified Training and Cortical Alignment

**Cortical Consistency Loss.** We maximize alignment of the cortex and subcortical structures between a pair of images while maintaining a smooth map.

We discretize the proposed consistency loss in Eq. (3), and use it to jointly train the volumetric and spherical registration networks. The consistency loss penalizes disagreement between the predicted 3D displacement on the cortex, and the 3D displacement computed from the predicted 2D warp:

$$\mathcal{L}_{cons}(\varphi, \psi, \mathcal{C}_1) = \frac{1}{N_v} \sum_{\boldsymbol{v} \in \mathcal{C}_1} f\left(\varphi\left(\boldsymbol{v}\right), \pi^{-1}\left(\psi\left(\pi\left(\boldsymbol{v}\right)\right)\right)\right), \tag{4}$$

where $N_v$ is the number of vertices in $\mathcal{C}_1$, $\pi(\boldsymbol{v})$ is the projection of $\boldsymbol{v}$ to the plane, and $f(\cdot, \cdot)$ is a similarity metric; we use the mean squared error in our experiments. We implement $\pi^{-1}(\cdot)$ with trilinear interpolation over the mapped planar parameterizations of $\boldsymbol{v}$. Maps $\varphi$ and $\psi$ are the outputs of the 3D and 2D CNN, respectively.

**Image Similarity Loss.** Both networks are trained by minimizing an unsupervised loss that balances image-similarity and field regularity. The similarity loss on the volume matches MRI intensities and the loss on the sphere matches geometric descriptors. We pre-compute geometric descriptors on the cortical meshes, $g : \mathcal{C} \to \mathbb{R}^2$. As is standard in spherical registration, we use the sulcal depth and mean curvature (Fischl et al., 1999; Cheng et al., 2020). The similarity loss is $\mathcal{L}_{sim}(\varphi, \psi) = \mathcal{L}_{sim}\left(I_2, \varphi\left(I_1\right)\right) + \mathcal{L}_{sim}\left(g\left(\pi\left(\mathcal{S}_2\right)\right), \psi\left(g\left(\pi\left(\mathcal{S}_1\right)\right)\right); w\left(\rho\right)\right)$. Here, $w(\rho)$ is a weighting to correct for area distortion introduced by the stereographic projection. The similarity loss measures pairwise similarity between the fixed image and sphere $I_2$, $\mathcal{S}_2$ and the mapped image and sphere, $\varphi(I_1)$, $\psi(\mathcal{S}_1)$. We use the local normalized cross correlation function as our similarity measure (Avants et al., 2010).

**Regularization.** We regularize the displacement fields to be smooth to encourage a well-behaved map. Our regularization loss is $\mathcal{L}_{reg}(\varphi, \psi) = \|\nabla \varphi\|^2 + \|\nabla \psi\|^2$, where $\nabla \varphi$ is the Jacobian of map $\varphi$.

**Auxiliary structural loss.** The use of anatomical label maps during training of learning-based registration often improves substructure alignment (Balakrishnan et al., 2019). When segmentation maps are available for some pairs of images, we use this information to supervise by structural alignment. Let $\text{seg}[I_1]$ indicate the segmentation labelmap of the $K$ brain substructures for image $I_1$. We add an additional loss to align these structures, $\mathcal{L}_{struc}(\varphi, \psi) = \mathcal{L}_{struc}(\text{seg}[I_2], \varphi(\text{seg}[I_1]) + \mathcal{L}_{struc}(\pi(\text{seg}[\mathcal{S}_2]), \pi(\psi(\text{seg}[\mathcal{S}_2])))$. In practice, we use the soft Dice loss function, which captures volume overlap of the warped subcortical structures on one image warped to the second.

**Joint training loss.** Combining all loss terms provides our joint optimization problem:

$$\mathcal{L}(\varphi, \psi) = \mathcal{L}_{sim}(\varphi, \psi) + \gamma \mathcal{L}_{cons}(\varphi, \psi) + \lambda \mathcal{L}_{reg}(\varphi, \psi) + \kappa \mathcal{L}_{struc}(\varphi, \psi), \tag{5}$$

where $\lambda, \gamma$, and $\kappa$ are hyperparameters.

## 4 Experiments

We evaluate NeurAlign using 3D brain MRI scans from multiple datasets comprising healthy subjects and subjects with cognitive disorders. We compare our method against CVS, the state-of-the-art combined cortical surface-volume registration framework (Postelnicu et al., 2008) and against learning-based methods that emphasize structural alignment. We test the ability of our method to generalize to new datasets unseen in training.

### 4.1 EXPERIMENTAL SETUP

**Data.**  We train our model using two public 3D brain MRI datasets: OASIS-1 (Marcus et al., 2007) and ADNI (Mueller et al., 2005). Both datasets are obtained from large population studies focused on Alzheimer's disease (AD). OASIS-1 (Marcus et al., 2007) includes T1-weighted (T1w) scans of 416 subjects aged 18-96. We use a random subset of 100 ADNI subjects aged 56–90, half of whom were diagnosed with AD. For each ADNI subject, we use data from two sessions, two years apart.

For each dataset, we hold out 20% of the subjects for testing, stratified by gender, age, and disease state. The remaining subjects are split into 85% for training and 15% for validation using the same stratification. In training, we randomly sample pairs from both datasets. In our test set, we create 123 non-overlapping pairs for registration: 83 pairs aligning OASIS-1 and OASIS-1, 20 pairs aligning ADNI and ADNI, and 20 pairs aligning OASIS-1 and ADNI.

**Held-out datasets.** We use two additional held-out datasets for testing model generalization. The IXI dataset contains MRI scans from healthy subjects acquired in 3 London hospitals (IXI Consortium). We randomly select T1w scans from 115 subjects for evaluation. We also hold out 80 T1w scans from the Mindboggle-101 dataset (Klein et al., 2017), a collection of manually labeled brain MRI volumes and surfaces. We exclude all OASIS-1 subjects from the Mindboggle dataset.

All evaluations are performed on T1w MRI, which is the modality that enables reliable cortical surface extraction. T1w MRI has the strongest gray-white matter contrast and supports reliable delineation of white and pial surfaces (Fischl et al., 1999; Fischl, 2012; Hoopes et al., 2022).

**Processing.** We use FreeSurfer (Fischl, 2012) to perform all image preprocessing. FreeSurfer is an open-source, widely used toolkit in neuroimaging that supports cortical surface analysis, longitudinal workflows, and population-scale MRI studies. We first perform affine alignment of each image to the Talairach template (Talairach & Tournoux, 1990). We crop all images to a roughly 10 mm margin around the brain. To prepare our training data, we generate anatomical segmentations, white matter (internal) and pial (external) cortical surface reconstructions, and spherically inflated surfaces with parameterized curvature maps. For evaluation, we use both volumetric segmentations of subcortical brain regions, computed on the volume, as well as parcellations of the cortical ribbon, computed on the surface mesh. Surface parcellations are rasterized as volumetric segmentations. We emphasize that our method does not require surface meshes, spheres, nor segmentation labelmaps at inference.

**Implementation Details.**  We train using the Adam optimizer (Kingma & Ba, 2014) with a learning rate of $10^{-4}$. We set the field regularization hyperparameter to $\lambda = 1.0$, the segmentation loss weight to $\kappa = 10.0$, and the cortical consistency weight to $\gamma = 0.05$. These were chosen using a grid search over the validation set. We train for $300,000$ iterations, using the best performing model on the validation set. All models are trained on a single $A100$ GPU. All CVS experiments are done on an Intel(R) Xeon(R) Gold 5218 CPU, as there is no GPU implementation.

**Baselines.**  We select state-of-the-art baselines that emphasize structural alignment. We evaluate CVS (Postelnicu et al., 2008; Zöllei et al., 2010), the most widely used joint volume-cortical alignment registration method, which is integrated in the FreeSurfer software suite. CVS performs sequential spherical to volumetric alignment using a biomechanical model and image intensity registration.

We evaluate uGradICON (Tian et al., 2024), a modern foundation model for medical image registration that generalizes well to unseen data. UniGradICON employs test-time optimization to improve pairwise-registration accuracy. We also use the uGradICON-seg variant that performs loss function masking using the segmentation labelmaps to improve cortical structure alignment.

We test SynthMorph (Hoffmann et al., 2023; 2024), a state-of-the-art registration tool trained exclusively on images synthesized from label maps. Since this tool optimized the overlap of standard FreeSurfer labels in training, which include only two large cortical structures, we also retrain it with extended label sets, optimizing overlap of the "a2009s" set of 196 labels that include finer Destrieux parcels of the cortex (Destrieux et al., 2010) (SynthMorph-wm).

We evaluate VoxelMorph (Dalca et al., 2019b), an unsupervised diffeomorphic registration model. We use the same size UNet as our model and train on our training set. We also evaluate FireANTs (Jena et al., 2025), a library for fast medical image registration using Riemannian optimization. We use the SyN transformation for symmetric diffeomorphic registration.

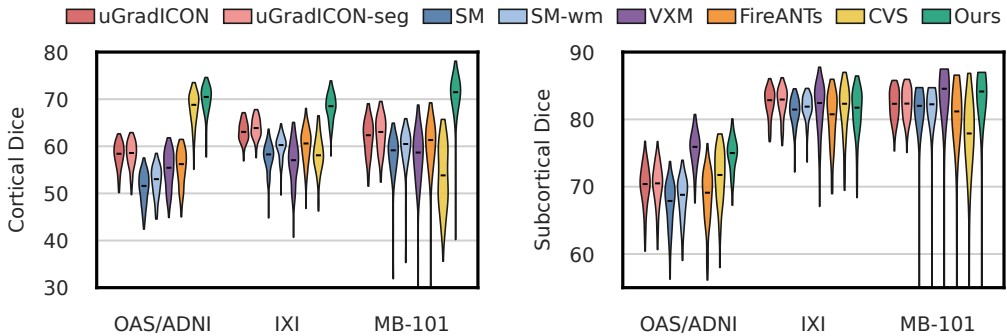

Figure 2: Violin plot of distributions of Dice score across all subjects in each dataset. Our method (blue) consistently achieves the highest mean Dice for cortical structures, with mass centered near the mean. We preserve subcortical performance while delivering much stronger cortical alignment.

**Evaluation.** To evaluate registration accuracy, we compute Dice scores between anatomical structures in the warped and target images. We report overlap for 43 subcortical structures and for cortical regions obtained by parcellating the cortical ribbon into 34 regions per hemisphere. We assess field regularity and topology by computing the determinant of the Jacobian of the map, $\det J_\varphi(p) = \det(\nabla\varphi(p))$ at each voxel $p$. Locations where $\det J_\varphi(p) < 0$ represent folded regions. We report the percentage of folds (% folds). We measure the smoothness of the warp fields using the standard deviation of $\log|\det J|$ (SDlogJ) (Leow et al., 2007).

## 4.2 RESULTS

Table 1 and Figure 2 present results on the 3 datasets. Our method, NeurAlign, consistently achieves the highest cortical Dice score, often by a large margin. For example, it improves by up to 7 Dice points on the held-out Mindboggle dataset. NeurAlign also achieves the highest subcortical Dice score on OASIS-1 & ADNI and Mindboggle. We obtain slightly lower subcortical Dice on IXI (1.5 points at worse), though the difference is not substantial. Across all datasets, NeurAlign achieves statistically better cortical Dice scores ($p < 0.01$) and in many cases a substantial increase (up to 7.5 points over the next best method). We also outperform baselines on whole cerebral cortex alignment (see Suppl. A.2). Finally, NeurAlign produces regular deformation fields with negligible folding. These results demonstrate that we are able to greatly improve cortical structure alignment while maintaining strong performance in substructure alignment and field regularity.

CVS was the slowest method, requiring $2.5 \pm 0.5$ hours per pair. uGradICON required minutes to perform test-time optimization. NeurAlign and all other baselines converged in milliseconds.

Figure 3 visualizes example registrations and corresponding warps. Our method yields accurate alignments with smooth and regular deformation fields.

## 4.3 ABLATIONS

To study important aspects of NeurAlign, we conduct two ablation experiments. In the first, we quantify the effect of different model components: label map supervision in training and including the spherical alignment loss. In the second, we quantify the effect of $\kappa$, the Dice loss weight parameter.

**Results.** Table 2 presents ablations of model components, trained with $\kappa = 1.0$ and $\gamma = 0.05$, on the IXI dataset. We train a base VoxelMorph model (Dalca et al., 2019a), called `Base`, and extensions testing components of NeurAlign. We compare the performance when training with supervision of segmentations of subcortical structures, `Base+Dice(subcort)`, training with segmentations of all structures, `Base+Dice(all)`, training with only the spherical consistency loss, `Base+Sphere`, and training with all proposed components, `Base+Dice(all)+Sphere`. We selected $\kappa = 1.0$, as it achieved the best performance for Base+Dice(subcort).

Table 1: Performance across three datasets. The proposed method (NeurAlign) substantially improves cortical dice and consistently matches or improves subcortical Dice, and produces smooth deformation fields with negligible folding. Mean $\pm$ standard deviation across subjects are indicated.

| Dataset | Method | Dice ($\uparrow$) cortical | Dice ($\uparrow$) subcortical | % folds ($\downarrow$) | SD log det J ($\downarrow$) |
|---|---|---|---|---|---|
| OASIS-1 & ADNI | uGradICON | $0.58 \pm 0.028$ | $0.701 \pm 0.032$ | $0.647 \pm 0.105$ | $0.408 \pm 0.127$ |
| | uGradICON-seg | $0.583 \pm 0.028$ | $0.701 \pm 0.031$ | $0.688 \pm 0.101$ | $0.41 \pm 0.126$ |
| | SynthMorph | $0.511 \pm 0.033$ | $0.673 \pm 0.034$ | $\mathbf{0.0 \pm 0.0}$ | $0.197 \pm 0.1$ |
| | SynthMorph-wm | $0.525 \pm 0.032$ | $0.682 \pm 0.031$ | $0.0 \pm 0.0$ | $\mathbf{0.188 \pm 0.1}$ |
| | VoxelMorph | $0.551 \pm 0.039$ | $\mathbf{0.756 \pm 0.026}$ | $0.001 \pm 0.001$ | $0.479 \pm 0.233$ |
| | FireANTs | $0.554 \pm 0.038$ | $0.684 \pm 0.041$ | $1.319 \pm 0.171$ | $0.5 \pm 0.137$ |
| | CVS | $0.681 \pm 0.035$ | $0.715 \pm 0.041$ | $1.73 \pm 0.321$ | $8.737 \pm 3.475$ |
| | **NeurAlign (Ours)** | $\mathbf{0.698 \pm 0.032}$ | $0.747 \pm 0.026$ | $0.169 \pm 0.037$ | $0.829 \pm 0.281$ |
| IXI | uGradICON | $0.631 \pm 0.021$ | $0.824 \pm 0.022$ | $0.489 \pm 0.1$ | $0.382 \pm 0.121$ |
| | uGradICON-seg | $0.639 \pm 0.021$ | $\mathbf{0.825 \pm 0.022}$ | $0.602 \pm 0.109$ | $0.399 \pm 0.125$ |
| | SynthMorph | $0.577 \pm 0.03$ | $0.811 \pm 0.021$ | $0.0 \pm 0.0$ | $0.177 \pm 0.069$ |
| | SynthMorph-wm | $0.599 \pm 0.025$ | $0.817 \pm 0.019$ | $\mathbf{0.0 \pm 0.0}$ | $\mathbf{0.167 \pm 0.064}$ |
| | VoxelMorph | $0.566 \pm 0.045$ | $0.816 \pm 0.038$ | $0.001 \pm 0.001$ | $0.465 \pm 0.17$ |
| | FireANTs | $0.601 \pm 0.037$ | $0.8 \pm 0.04$ | $1.382 \pm 0.183$ | $0.501 \pm 0.133$ |
| | CVS | $0.582 \pm 0.039$ | $0.814 \pm 0.038$ | $1.865 \pm 0.382$ | $7.591 \pm 4.115$ |
| | **NeurAlign (Ours)** | $\mathbf{0.683 \pm 0.029}$ | $0.81 \pm 0.036$ | $0.081 \pm 0.023$ | $0.712 \pm 0.24$ |
| Mindboggle | uGradICON | $0.618 \pm 0.038$ | $0.821 \pm 0.023$ | $0.85 \pm 0.404$ | $0.436 \pm 0.199$ |
| | uGradICON-seg | $0.626 \pm 0.038$ | $0.822 \pm 0.023$ | $0.958 \pm 0.4$ | $0.445 \pm 0.188$ |
| | SynthMorph | $0.577 \pm 0.054$ | $0.806 \pm 0.065$ | $0.0 \pm 0.0$ | $0.185 \pm 0.071$ |
| | SynthMorph-wm | $0.596 \pm 0.048$ | $0.811 \pm 0.059$ | $\mathbf{0.0 \pm 0.0}$ | $\mathbf{0.173 \pm 0.062}$ |
| | VoxelMorph | $0.573 \pm 0.072$ | $\mathbf{0.827 \pm 0.09}$ | $0.002 \pm 0.002$ | $0.51 \pm 0.245$ |
| | FireANTs | $0.597 \pm 0.07$ | $0.79 \pm 0.09$ | $1.604 \pm 0.347$ | $0.53 \pm 0.169$ |
| | CVS | $0.535 \pm 0.07$ | $0.766 \pm 0.081$ | $2.164 \pm 0.652$ | $7.429 \pm 4.654$ |
| | **NeurAlign (Ours)** | $\mathbf{0.703 \pm 0.06}$ | $0.823 \pm 0.086$ | $0.174 \pm 0.034$ | $0.831 \pm 0.261$ |

As expected, training with subcortical supervision improved the Dice (subcortical) score compared to the Base model. However, interestingly, training with supervision of all brain structures did not lead to a substantial improvement in the Dice (cortical) score. Only when including our proposed spherical consistency loss (Base+Dice(all)+Sphere) did we observe a substantial increase in the Dice (cortical) score while maintaining the Dice (subcortical) score. Training with just the spherical loss without structural supervision did not improve results, indicating that we also require volumetric structure supervision. This ablation study demonstrates the benefit of our spherical-volumetric consistency constraint in obtain accurate alignment of fine-grained cortical structures.

We assess the impact of $\kappa$, the weighting hyperparameter for the Dice loss term (see Suppl. Table A.1, Fig. A.1). As expected, increasing $\kappa$ improves Dice performance at test time, yielding higher structural overlap. In some regions, however, this comes at the expense of deformation field regularity. Overall, setting $\kappa > 1$ produces models that outperform baselines across most metrics. Consistent with prior work (Dalca et al., 2019b; Hoopes et al., 2021), the optimal choice of $\kappa$ depends on the intended downstream application: for atlas-based segmentation propagation, higher Dice weighting is advantageous, whereas for longitudinal studies, smoother deformation fields may be preferable. Hypernetworks can be considered to select hyperparameters at test-time (Hoopes et al., 2021).

Table 2: Ablations over training settings on IXI. The proposed spherical alignment consistency loss is necessary to obtain high cortical structure alignment while maintaining strong subcortical alignment.

| Ablation | Dice ($\uparrow$) cortical | Dice ($\uparrow$) subcortical | % folds ($\downarrow$) | SD log det J ($\downarrow$) |
|---|---|---|---|---|
| Base | $0.582 \pm 0.045$ | $0.789 \pm 0.049$ | $0.003 \pm 0.003$ | $0.446 \pm 0.172$ |
| Base+Dice (subcort) | $0.553 \pm 0.045$ | $\mathbf{0.815 \pm 0.038}$ | $0.002 \pm 0.001$ | $0.494 \pm 0.185$ |
| Base+Dice (all) | $0.562 \pm 0.044$ | $0.812 \pm 0.041$ | $\mathbf{0.001 \pm 0.001}$ | $0.462 \pm 0.168$ |
| Base+Sphere | $0.56 \pm 0.03$ | $0.736 \pm 0.047$ | $0.004 \pm 0.002$ | $\mathbf{0.421 \pm 0.152}$ |
| Base+Dice(all)+Sphere | $\mathbf{0.633 \pm 0.028}$ | $0.799 \pm 0.038$ | $0.006 \pm 0.003$ | $0.432 \pm 0.137$ |

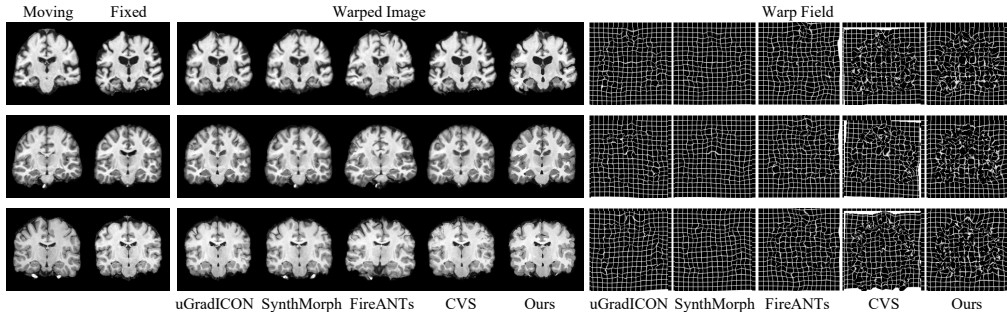

Figure 3: Example registrations for three image pairs. Columns show the moving and fixed images, followed by warped images produced by uGradICON-seg, SynthMorph-wm, FireANTs, CVS, and our method. Corresponding deformation fields are visualized on the right. Our method (NeurAlign) yields accurate alignments with smooth and regular warp fields.

## 5 DISCUSSION

**Limitations and Future Work.**    We presented NeurAlign, a model for unified cortical and subcortical neuroimage registration, evaluated primarily on adult T1w MRI from healthy and Alzheimer's disease populations. As with any diffeomorphic registration framework, NeurAlign cannot accurately accommodate topology-altering pathologies. This limitation can be partially mitigated by incorporating pathology masks during training (Brett et al., 2001). It also remains unclear how well the model can generalize to lower-quality clinical scans or to pediatric populations. Extending NeurAlign to these more challenging settings is an important direction for future work.

Training our model requires cortical surface extraction and spherical inflation. This preprocessing pipeline may fail on challenging scans. A simple approach for quality control could be to discard failed cases, for example surfaces with many self-intersecting faces, or to use robust, learning-based extraction methods (Hoopes et al., 2022; Bongratz et al., 2022). In practice, we did not observe any failures on our training data.

Our model was also only tested on the T1w modality. T1w is the most widely used imaging modality for structural imaging studies (Bhalerao et al., 2024), and it is currently the primary modality from which cortical surfaces can be reliably reconstructed (Fischl, 2012; Hoopes et al., 2022) An interesting future direction is extending NeurAlign to multimodal acquisitions. Additional contrasts such as T2w, FLAIR, or DWI could provide complementary information for subcortical tissue segmentation or characterization. Incorporating these modalities would require multimodal training with all contrasts aligned to the T1w image so that the cortical surfaces remain consistent across channels. Importantly, at inference time, the surfaces would no longer be required, potentially reducing the current dependence on T1w imaging for surface extraction.

A benefit of our formulation is that it can be easily extended to include additional structures with genus-0 topology. For example, this can improve alignment of the hippocampus by providing additional supervision through surface alignment. Further, our consistency loss in Eq. (3) can be adopted for non-spherical surface registration, implicit surfaces, or to perform cortical alignment directly on the surface mesh using frameworks such as Deep Functional Maps (Donati et al., 2020). Representations beyond spheres can be used, provided there is a one-to-one map to the cortical mesh.

**Conclusion.**    We introduced NeurAlign, a deformable registration method that achieves accurate cortical and subcortical alignment. NeurAlign parameterizes the deformation field through coupled spherical and volumetric alignment, promoting topological consistency on the cortex.

Our model outperforms state-of-the-art deep learning and foundational models trained or fine-tuned using cortical segmentation label maps. Compared to CVS, the most widely used method in large-scale neuroimaging pipelines, our method achieves substantially better alignment and deformation field regularity on held-out datasets, while running orders of magnitude faster. Further, our method does not require additional inputs (cortical meshes, segmentations) as inputs. This positions our method as a practical alternative to CVS for many neuroimaging applications.

REPRODUCIBILITY STATEMENT

Our model is summarized in Section 3.3, including descriptions of the 3D and 2D models used and loss functions in training. We provide implementation details including all hyperparameters used, and data processing details in Section 4.1. Complete code for full reproducibility will be made public upon acceptance.

ACKNOWLEDGMENTS

We thank Neel Dey for helpful discussions. Support for this research was provided in part by Quanta Computer Inc., the BRAIN Initiative Cell Atlas Network (BICAN) grants U01MH117023, UM1MH134812 and UM1MH130981, the Brain Initiative Brain Connects consortium (U01NS132181, 1UM1NS132358-01), the National Institute for Biomedical Imaging and Bioengineering (R01 EB033773, 2R01EB023281, R21EB018907, R01EB019956, P41EB030006), the National Institute on Aging (R21AG082082, 1R01AG064027, R01AG016495, 1R01AG070988), the National Institute of Mental Health (UM1MH130981, R01 MH123195, R01 MH121885, 1RF1MH123195), the National Institute for Neurological Disorders and Stroke , (1U24NS135561-01, R01NS070963, 2R01NS083534, R01NS105820, R25NS125599), the National Insititute for Child Health and Human Development (R01HD109436, R00HD101553), and was made possible by the resources provided by Shared Instrumentation Grants 1S10RR023401, 1S10RR019307, and 1S10RR023043. Additional support was provided by the NIH Blueprint for Neuroscience Research (5U01-MH093765), part of the multi-institutional Human Connectome Project. Parts of this work were also supported by the ERC Starting Grant 758800 (EXPROTEA), ERC Consolidator Grant 101087347 (VEGA), as well as gifts from Ansys and Adobe Research. Much of the computation resources required for this research was performed on computational hardware generously provided by the Massachusetts Life Sciences Center (https://www.masslifesciences.com/). In addition, BF is an advisor to DeepHealth, a company whose medical pursuits focus on medical imaging and measurement technologies. AD is an advisor to Radence and DeepHealth. BF's and AD's interests were reviewed and are managed by Massachusetts General Hospital and Mass. General Brigham in accordance with their conflict of interest policies.

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

# A  Supplementary Material

## A.1  Ablation over Dice Loss Weight

In this section, we conduct ablations over the Dice loss weight $\kappa$. We use the IXI dataset and a sphere loss weight $\gamma = 0.05$. As demonstrated on Table 2, the inclusion of the soft Dice loss on cortical and subcortical structures in combination with our proposed spherical alignment loss is necessary to produce accurate cortical and subcortical alignments. However, this comes at the cost of field regularity. Table A.1 presents the results. Consistent increases in Dice (cortical) are observed with increasing $\kappa$, but Dice (subcortical) starts to saturate for $\kappa > 5$. This comes at the expense of field regularity, as measured by the % of folds and SD$\log \det J$.

Figure A.1 presents example images and warps varying $\kappa$. As expected, deformation fields become more irregular with increasing $\kappa$, though presented ranges are reasonable. The optimal choice of $\kappa$ depends on the intended downstream application. The user has control over the desired strength of structural alignment over field regularity and smoothness. We note however that our model produces accurate matchings with regular fields for a large range of $\kappa$ values.

Table A.1: Ablations over Dice loss weight $\kappa$ on the IXI dataset.

| Dice loss weight | Dice (↑) cortical | Dice (↑) subcortical | % folds (↓) | SD $\log \det$ J (↓) |
|---|---|---|---|---|
| $\kappa = 0.5$ | $0.613 \pm 0.029$ | $0.787 \pm 0.041$ | $0.014 \pm 0.005$ | $0.459 \pm 0.15$ |
| $\kappa = 1.0$ | $0.633 \pm 0.028$ | $0.799 \pm 0.038$ | $\mathbf{0.006 \pm 0.003}$ | $\mathbf{0.432 \pm 0.137}$ |
| $\kappa = 2.0$ | $0.637 \pm 0.03$ | $0.793 \pm 0.037$ | $0.016 \pm 0.006$ | $0.462 \pm 0.141$ |
| $\kappa = 5$ | $0.653 \pm 0.029$ | $0.796 \pm 0.039$ | $0.046 \pm 0.015$ | $0.597 \pm 0.188$ |
| $\kappa = 10$ | $0.683 \pm 0.029$ | $\mathbf{0.81 \pm 0.036}$ | $0.081 \pm 0.023$ | $0.712 \pm 0.24$ |
| $\kappa = 50$ | $\mathbf{0.691 \pm 0.028}$ | $0.799 \pm 0.038$ | $0.848 \pm 0.135$ | $1.294 \pm 0.414$ |

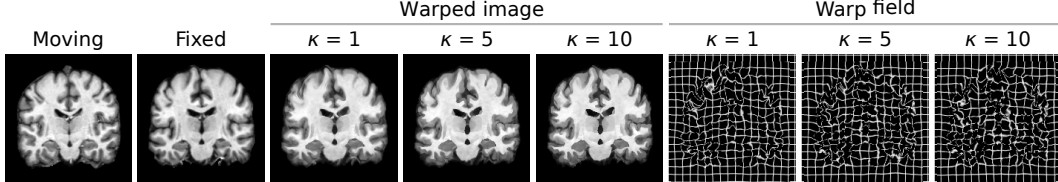

Figure A.1: Example registrations for one image while varying $\kappa$. Columns show the moving and fixed images, followed by warped images and corresponding deformation fields are visualized on the right. Deformation fields become more irregular with increasing $\kappa$, however all demonstrated values produce accurate and smooth alignments.

## A.2  Whole Cerebral Cortex Alignment Performance

Our evaluation on Table 1 and Fig. 2 focused on quantifying performance of aligning cortical parcellations within the cerebral cortex. We hypothesized that the spherical alignment signal will be most important for aligning the homologous folds as the image intensity information is unable to capture the functional regions within the cortex. To test this hypothesis, we compared Dice score in aligning cortical parcellations (Dice cortical) with Dice score aligning the entire cerebral cortex (Dice CC), which groups all parcellations as one class. Table A.2 presents the results.

On two of the three datasets, our model achieves the highest Dice (CC) performance, while CVS outperforms our method by $1.2$ Dice points on the OASIS-1 & ADNI test set. Interestingly, all baseline methods show improved performance in aligning the cerebral cortex and differences across baselines are smaller. This indicates that purely volumetric approaches may be sufficient at capturing the whole cerebral cortex structure. However, when trying to align the cortical parcellations, the benefit of our formulation is immediate, as we obtain $1.9 - 7.5$ point Dice increases over the next best

Table A.2: Performance of all methods across three datasets, focusing on structural alignment on the graymatter. Dice (CC) indicates the dice score in aligning the entire cerebral cortex, while Dice cortical refers to the parcellated substructures within the cerebral cortex. The benefit of our method in aligning substructures within the cortex (Dice cortical) is immediately apparent, as we consistently achieve the highest score, including up to a 7.5 dice score improvement over the next method. When considering the cerebral cortex as one structure, performance differences decrease, though our method still performs the strongest on 2 of the 3 datasets. The benefit of the spherical registration is immediately apparent as it is necessary to align functional regions in the cortex where image-only approaches fail. Mean $\pm$ standard deviation across subjects are indicated.

| Dataset | Method | Dice (CC) ($\uparrow$) left | Dice (CC) ($\uparrow$) right | Dice ($\uparrow$) cortical | % folds ($\downarrow$) |
|---|---|---|---|---|---|
| OASIS-1 & ADNI | uGradICON | $0.711 \pm 0.029$ | $0.71 \pm 0.026$ | $0.58 \pm 0.028$ | $0.647 \pm 0.105$ |
| | uGradICON-seg | $0.712 \pm 0.03$ | $0.711 \pm 0.027$ | $0.583 \pm 0.028$ | $0.688 \pm 0.101$ |
| | SynthMorph | $0.626 \pm 0.035$ | $0.628 \pm 0.032$ | $0.511 \pm 0.033$ | $0.0 \pm 0.0$ |
| | SynthMorph-wm | $0.617 \pm 0.035$ | $0.62 \pm 0.031$ | $0.525 \pm 0.032$ | $\mathbf{0.0 \pm 0.0}$ |
| | VoxelMorph | $0.72 \pm 0.031$ | $0.72 \pm 0.03$ | $0.551 \pm 0.039$ | $0.001 \pm 0.001$ |
| | FireANTs | $0.703 \pm 0.034$ | $0.703 \pm 0.033$ | $0.554 \pm 0.038$ | $1.319 \pm 0.171$ |
| | CVS | $\mathbf{0.777 \pm 0.03}$ | $\mathbf{0.775 \pm 0.03}$ | $0.681 \pm 0.035$ | $1.73 \pm 0.321$ |
| | **NeurAlign (Ours)** | $0.765 \pm 0.031$ | $0.764 \pm 0.025$ | $\mathbf{0.698 \pm 0.032}$ | $0.169 \pm 0.037$ |
| IXI | uGradICON | $0.74 \pm 0.02$ | $0.746 \pm 0.017$ | $0.631 \pm 0.021$ | $0.489 \pm 0.1$ |
| | uGradICON-seg | $0.749 \pm 0.02$ | $0.755 \pm 0.017$ | $0.639 \pm 0.021$ | $0.602 \pm 0.109$ |
| | SynthMorph | $0.691 \pm 0.02$ | $0.693 \pm 0.02$ | $0.577 \pm 0.03$ | $\mathbf{0.0 \pm 0.0}$ |
| | SynthMorph-wm | $0.683 \pm 0.021$ | $0.684 \pm 0.021$ | $0.599 \pm 0.025$ | $0.0 \pm 0.0$ |
| | VoxelMorph | $0.736 \pm 0.033$ | $0.737 \pm 0.032$ | $0.566 \pm 0.045$ | $0.001 \pm 0.001$ |
| | FireANTs | $0.753 \pm 0.025$ | $0.754 \pm 0.024$ | $0.601 \pm 0.037$ | $1.382 \pm 0.183$ |
| | CVS | $0.748 \pm 0.041$ | $0.754 \pm 0.036$ | $0.582 \pm 0.039$ | $1.865 \pm 0.382$ |
| | **NeurAlign (Ours)** | $\mathbf{0.758 \pm 0.023}$ | $\mathbf{0.758 \pm 0.021}$ | $\mathbf{0.683 \pm 0.029}$ | $0.081 \pm 0.023$ |
| Mindboggle | uGradICON | $0.722 \pm 0.051$ | $0.731 \pm 0.044$ | $0.618 \pm 0.038$ | $0.85 \pm 0.404$ |
| | uGradICON-seg | $0.73 \pm 0.049$ | $0.738 \pm 0.045$ | $0.626 \pm 0.038$ | $0.958 \pm 0.4$ |
| | SynthMorph | $0.697 \pm 0.03$ | $0.701 \pm 0.036$ | $0.577 \pm 0.054$ | $\mathbf{0.0 \pm 0.0}$ |
| | SynthMorph-wm | $0.685 \pm 0.031$ | $0.688 \pm 0.038$ | $0.596 \pm 0.048$ | $0.0 \pm 0.0$ |
| | VoxelMorph | $0.736 \pm 0.055$ | $0.744 \pm 0.055$ | $0.573 \pm 0.072$ | $0.002 \pm 0.002$ |
| | FireANTs | $0.742 \pm 0.05$ | $0.748 \pm 0.051$ | $0.597 \pm 0.07$ | $1.604 \pm 0.347$ |
| | CVS | $0.689 \pm 0.091$ | $0.693 \pm 0.088$ | $0.535 \pm 0.07$ | $2.164 \pm 0.652$ |
| | **NeurAlign (Ours)** | $\mathbf{0.782 \pm 0.04}$ | $\mathbf{0.785 \pm 0.041}$ | $\mathbf{0.703 \pm 0.06}$ | $0.173 \pm 0.034$ |

method across all datasets. The purely volumetric approaches show substantially lower performance than our method, as well.

