# OpenReview forum: "Unified Brain Surface and Volume Registration"
_ICLR.cc/2026/Conference — ICLR 2026 Poster_

### Official Review · Reviewer_L2kZ · 2025-10-29

**Soundness:** 3
**Presentation:** 3
**Contribution:** 3
**Rating:** 8
**Confidence:** 4

**Summary:**

This paper proposes a unified deep learning framework for brain MRI registration that jointly aligns cortical surfaces and volumetric subcortical structures using spherical + volumetric CNNs coupled by a consistency loss. The work aims to overcome limitations of existing methods (e.g., CVS) that treat the two domains sequentially.

**Strengths:**

Clear contribution: A unified model bridging spherical and volumetric registration is well-motivated.

Technical novelty: The cortical consistency loss connecting domains is elegant and addresses a real limitation in current pipelines.

Strong results: Substantial gains in cortical Dice (up to +7 points) on multiple datasets, with very fast inference vs. CVS.

Generalization: Tested on multiple held-out datasets including IXI and Mindboggle.

Solid ablation studies isolating effects of proposed components.

**Weaknesses:**

Dependence on spherical preprocessing at training: Requires surface extraction and inflation (FreeSurfer), which can be slow and may fail on challenging scans; more discussion on robustness would help.

Subcortical performance not universally better: Slight drop vs. uniGradICON/SynthMorph on IXI subcortical structures.

Limited modality / population scope: All training on adult T1-weighted datasets (OASIS/ADNI). No fetal/pediatric/clinical robustness experiments.

Ablations could explore more consistency-loss formulations (e.g., geodesic metrics on the sphere).

Clarity: Some math sections are dense; figure captions could better explain preprocessing needed.


No code availabel.

The paper is a bit dense I think more figure would help.

**Questions:**

Surface preprocessing dependency
Your method requires cortical meshes and spherical inflation during training. How sensitive is performance to inaccuracies or failures in this preprocessing? Could UCS be trained without spherical meshes (e.g., using synthetic or implicit surfaces)?

Inference on challenging scans
Since no surface information is used at inference, how well does the model perform on:

low-resolution or motion-corrupted clinical MRIs?

subjects with pathology altering cortical shape (e.g., tumors, resections)?
Any preliminary results or failure analyses?

Consistency loss mechanics
Could you elaborate on the interpolation needed to map spherical displacement back to 3D mesh vertices? How does sampling density on the sphere affect gradient stability?

Subcortical performance variability
Your method slightly underperforms baselines on IXI subcortical alignment.
Why might IXI behave differently? (acquisition style? age distribution?)
Would modest segmentation supervision balance this out?

Jacobian regularity vs. Dice trade-off
Ablations show increasing κ improves Dice but degrades regularity.
Do you have guidance on selecting κ for different applications (e.g., longitudinal studies)? Could adaptive balancing be learned?

Topology enforcement
While spherical alignment preserves cortical topology, volumetric deformation can still introduce folding. Would incorporating diffeomorphic constraints on the spherical path reduce the remaining ~0.1–0.2% folds?

Sphere parameterization distortion
Stereographic projection introduces sampling and area distortion.
How does this influence fine-scale folding alignment? Any alternatives considered (e.g., icosahedral CNNs)?

Training pairs
Registration direction is arbitrary. Did you augment by swapping moving/fixed pairs? Any asymmetry observed in deformation field quality?

Runtime comparison fairness
Since CVS requires spherical meshes at inference, did timing exclude preprocessing?
Could UCS runtime remain competitive if including preprocessing of spheres during training on real pipelines?

Scalability to groupwise registration
Could your framework support joint template estimation or atlas building?
What changes would be required to extend UCS to multi-subject alignment?

---

> ### Author Response · Authors · 2025-11-26
>
> We thank the reviewer for outlining the strengths of our method including a clear, and well-motivated contribution with technical novelty and strong experimental results that generalize to multiple held-out datasets.
>
> > Dependence on spherical preprocessing at training: Requires surface extraction and inflation (FreeSurfer), which can be slow and may fail on challenging scans; more discussion on robustness would help.
>
> > Surface preprocessing dependency Your method requires cortical meshes and spherical inflation during training. How sensitive is performance to inaccuracies or failures in this preprocessing?
>
> Thank you for bringing up this point. You are correct that surface extraction may fail on challenging scans. In our experiments, FreeSurfer succeeded in all cases in our training set. Occasional failures could be manually handled by removing them from training. A strength of our model is that it does not require cortical meshes at inference time, alleviating any pitfalls of needing to extract meshes at inference.
>
> In practice, we found that our model could successfully generalize to two held-out datasets in inference, suggesting that the model is robust.
>
> Furthermore, our formulation can work with any cortical surface extraction and inflation method. Aside from FreeSurfer, users could also use existing deep learning tools that have demonstrated increased robustness, such as vox2cortex (Bongratz et al., 2002) and TopoFit (Hoopes et al., 2022).
>
> We add the following text to Section 5 Discussion to explicitly address this:
>
> *Training our model requires cortical surface extraction and spherical inflation. This preprocessing pipeline may fail on challenging scans. A simple approach for quality control could be to simply discard failed cases, for example surfaces with many self-intersecting faces, or to use robust, learning-based extraction methods (Hoopes et al., 2022; Bongratz et al., 2022). In practice, we did not observe any failures on our training data.*
>
> > Subcortical performance not universally better: Slight drop vs. uniGradICON/SynthMorph on IXI subcortical structures.
>
> You are correct in pointing this out. In the worst case, UCS underperforms uniGradICON-seg by 1.5 Dice points in subcortical alignment, and the difference is small. UCS however outperforms in cortical Dice alignment by 8-points compared with SynthMorph and 4.5 points compared with uGradICON-seg, both substantial and significant improvements.  uGradICON-seg also **requires** the input segmentation masks at inference, which is often not available. Depending on the dataset and user goals, tuning the $\kappa$ hyperparameter can further improve the subcortical alignment and hence Dice score. These results demonstrate that UCS substantially improves cortical alignment while maintaining state-of-the-art performance in subcortical alignment.
>
> We add text to Section 4.2 to describe this difference:
>
> *We obtain slightly lower subcortical Dice on IXI (1.5 points at worst), though the difference is not substantial. Across all datasets, we achieve statistically better cortical Dice scores (p < 0.01) and in many cases a substantial increase (up to 7.5 points over the next best method).*
>
> > Limited modality / population scope: All training on adult T1-weighted datasets (OASIS/ADNI). No fetal/pediatric/clinical robustness experiments.
>
> The focus of this work was on adult T1w MRI data. Pediatric and infant datasets are challenging in practice: fetal data does not have substantial myelination to require cortical surface alignment, and infant and early developmental brains have large differences in cortical folding patterns, tissue contrasts, and anatomical variability from adult brain MRI. Further, infant data does not have the detailed cortical parcellations that we are interested in. Consequently, we chose to focus on adult brains for this study, and leave infant scans for future work.
>
> As 1mm isotropic T1w MRI is the standard modality and resolution for structural neuroimaging studies, we focused our training and evaluation on this type of data to demonstrate that our method can replace existing tools, such as CVS, in standard neuroimaging pipelines. We further demonstrate generalization to two additional datasets unseen in training. Our experimental evaluation covers a large age range and includes healthy patients, and patients with cognitive impairment and Alzheimer’s disease.
>
> We agree that it would be beneficial for future work to include clinical scans in our analysis, and we have mentioned this in Section 5 Discussion, subsection Limitations and Future Work. We agree that it would be great to compare with clinical quality scans. If the reviewer has suggestions on open datasets, we would be happy to include these comparisons in our revised paper.
>
> We are currently trying to obtain approval to use low-quality post mortem MRI scans from an ongoing study. If such approval is obtained, we will include qualitative results.

---

> > ### Author Response · Authors · 2025-11-26
> >
> > > Ablations could explore more consistency-loss formulations (e.g., geodesic metrics on the sphere).
> >
> > Thank you for making this suggestion. Geodesic metrics on the sphere is a great idea. We had thought of how to formulate this, however, we found issues on the 3D (volumetric) side of this loss. Computing differences in spherical geodesics would require projecting the 3D volumetric displacement of the mesh onto the sphere. This is likely to lead to many vertices being projected to similar locations on the sphere, and breaking bijectivity of the map. As the volumetric displacement field often maps cortical vertices to locations *off* the target cortex, the projection step may introduce a lot of distortion and obscure the mapping.
> >
> > We would be happy to add additional ablations testing different consistency-loss formulations, if you have suggestions on how to deal with the projection issue, or have additional loss function suggestions, we will happily include them.
> >
> > We add the following text to Section 5 Discussion, subsection Limitations and Future Work
> >
> > *Our consistency loss in Eq. (3) can be adopted for non-spherical surface registration, implicit surfaces, or to perform cortical alignment directly on the surface mesh using existing frameworks such as Deep Functional Maps (Donati et al.,2020).*
> >
> > > Clarity: Some math sections are dense; figure captions could better explain preprocessing needed.
> >
> > Thank you for raising this issue. We have added additional notation to Figure 1 and expanded the caption to discuss the preprocessing steps needed.
> >
> > > No code available.
> >
> > We will release code upon acceptance of the paper.
> >
> > ## Questions
> > Thank you for raising several interesting questions. We address them below.
> >
> > > Could UCS be trained without spherical meshes (e.g., using synthetic or implicit surfaces)?
> >
> > In principle, our method could be extended to train with other types of surface representations, though this would require modifying architectures and the training framework. The main term coupling the spherical and volumetric registration is in Equation (3). One could presumably use alignment losses on implicit surfaces or meshes directly. For example, the spherical registration network could be replaced by a Deep Functional maps network that learns a pointwise correspondence directly on the cortical surface mesh. Provided there is a one-to-one map between the cortical surface mesh and the surface representation used in registration (e.g. sphere, implicit surface, or mesh itself), the framework is extendable to other representations. We would note however that based on past work, the spherical representation is the best one as it guarantees the topology of the optimization is constrained to the cortical surface.
> >
> > We add the following text to the future work section:
> >
> > *Our consistency loss in Eq. (3) can be adopted for non-spherical surface registration, implicit surfaces, or to perform cortical alignment directly on the surface mesh using existing frameworks such as Deep Functional Maps (Donati et al.,2020). Cortical surface representations beyond spheres can be used, provided there is a one-to-one map to the cortical surface mesh.*
> >
> > > Inference on challenging scans Since no surface information is used at inference, how well does the model perform on:
> > > - subjects with pathology altering cortical shape (e.g., tumors, resections)? Any preliminary results or failure analyses?
> > > - low-resolution or motion-corrupted clinical MRIs?
> >
> > Our method would not work on pathology that alters cortical surface topology (e.g. tumors, resections). We assume a genus-0 topology and learn diffeomorphic transformations. Diffeomorphism would break down in this case. In general, this is a challenging problem in registration, and often requires that both the source and target image have the same pathology. One approach would be to mask the tumor region and perform the registration with 0-masking in the loss function.
> >
> > We are currently in the process of obtaining permission for a low-resolution, clinical MRI but have not yet received approval to include the paper. We hope to do so by the end of the rebuttal period. If you have specific datasets in mind to evaluate our method on low-resolution or motion-corrupted MRI, we would be more than happy to do so.
> >
> > To address these limitations, we have modified the text in Section 5 Discussion:
> >
> > *We presented UCS, a model for unified cortical and subcortical neuroimage registration, evaluated primarily on adult T1w MRI from healthy and Alzheimer’s populations. As with any diffeomorphic registration framework, \method{} cannot accurately accommodate topology-altering pathologies. This limitation can be partially mitigated by incorporating pathology masks during training (Brett et al., 2001). It also remains unclear how well the model can generalize to lower-quality clinical scans or to pediatric populations. Extending UCS to these more challenging settings is an important direction for future work.*

---

> > > ### Author Response · Authors · 2025-11-26
> > >
> > > > Consistency loss mechanics Could you elaborate on the interpolation needed to map spherical displacement back to 3D mesh vertices? How does sampling density on the sphere affect gradient stability?
> > >
> > > Each 3D coordinate $(x_k,y_k,z_k)$ has a corresponding coordinate on the sphere $(\theta_k,\phi_k)$ (i.e. a one-to-one map). The spherical parameterization is a stereographic projection function that maps the spherical coordinates to a discrete 2D grid. For example, grid coordinates $(i,j) = \pi((\theta_k,\phi_k))$. We can similarly use the same projection function as a lookup table to assign 3D coordinates to each 2D grid location, since each spherical coordinate $(\theta,\phi)$ has a corresponding 3D coordinate $(x,y,z)$. We also precompute the inverse mapping: every grid location $(i,j)$ is associated with its corresponding 3D surface coordinate via lookup table $L(i,j)$.
> > >
> > > The 2D spherical UNet learns a displacement function $\psi$ that operates on the 2D grid. A displaced grid point $(i*,j*)=(i,j)+\psi(i,j)$ is then mapped back to 3D space by bilinear interpolation of the lookup table $L$. This yields the corresponding displaced 3D point on the cortical surface. This forward pass is fully differentiable because both the projection $\pi$ and bilinear sampling $L$ are fixed linear operators.
> > >
> > > We added the following text to Section 3.3.1 Unified Training and Cortical Alignment, subsection Cortical Consistency Loss:
> > >
> > > *We implement $\pi^{-1}(\cdot)$ with trilinear interpolation over the mapped planar parameterizations of $\mathbf v$.*
> > >
> > > >Gradient stability and sampling density
> > >
> > > Thank you for raising this point. The gradient of the consistency loss with respect to the spherical displacement simply backpropagates through the bilinear sampling of the lookup table $L$, which is standard for spatial-transformer layers. Consequently, we do not expect to see gradient instability from sampling density alone.
> > >
> > > However, sampling density affects the spatial resolution of the displacement field. The spherical sampling density determines how finely the displacement field $\psi$ is represented on the sphere. A coarser grid implies that each grid cell covers a larger region on the cortical surface after projection; therefore, any variation in surface geometry within that region is aggregated when we bilinearly sample $L$. This leads to a smoother, lower-frequency representation of the spherical deformation, limiting how precisely the spherical branch can match fine-scale surface details.
> > >
> > > Conversely, a denser spherical grid results in smaller spatial steps in (i,j), meaning the map captures more localized geometric variation. A higher resolution more accurately captures the fine geometric features of the cortical mesh.
> > >
> > > In the paper, we chose roughly a one-to-one resolution between the number of grid points and mesh vertices. This was chosen to accurately represent the cortical surface (and sphere) in a 2D plane.
> > >
> > > However, the projection to the 2D plane does introduce additional complications to account for, regarding sampling distortion at the poles. The spherical parameterization introduces denser sampling for higher latitude regions, i.e. the north and south poles are stretched the width of the image. To account for this, we perform distortion correction in each loss function by weighing samples by $\sin(\theta)$, where $\theta$ is the polar angle describing the elevation (Cheng et al., 2020; Li et al., 2024).
> > >
> > > We have added text to Section 3.3 Discrete Model, subsection Spherical Registration Network:
> > >
> > > *To account for the nonuniform sampling of the spherical parameterization, where regions near the poles are stretched, we apply distortion correction in all surface-based losses by weighting samples by $\sin(\theta)$ (Cheng et al., 2020; Li et al., 2024), where $\theta$ is the elevation angle.*

---

> > > > ### Author Response · Authors · 2025-11-26
> > > >
> > > > >Subcortical performance variability Your method slightly underperforms baselines on IXI subcortical alignment. Why might IXI behave differently? (acquisition style? age distribution?) Would modest segmentation supervision balance this out?
> > > >
> > > > Thank you for raising this point and suggesting ways to close this gap.
> > > >
> > > > On IXI, our method shows a small decrease in subcortical Dice (0.08-0.15) compared to uGradICON and SynthMorph, while achieving substantially higher cortical Dice (4.4–10.6 points). This is consistent with the current design of our model, which emphasizes accurate cortical alignment and a smooth, coupled deformation field. We also note that the IXI subcortical Dice differences between our method and the baselines are not statistically significant. Finally, uGradICON obtains 0.49% to 0.60% folded voxels compared to 0.08% with our method. uGradICON thus may be obtaining better Dice due to less field regularity enforcement. In practice, one is more likely to favor our method’s balance in substantially improved cortical alignment, slightly worse subcortical alignment, with smoother deformation fields.
> > > >
> > > > As you mentioned, modest supervision would likely balance this out. A straightforward extension would be to use a second hyperparameter on the subcortical structures. Similarly, including volume-weighted Dice loss term would likely aid in aligning subcortical structures, as the cortical structures are small in comparison. These modifications do not require architectural changes and would provide a direct mechanism to tighten subcortical alignment when desired. In practice, however, the observed IXI differences are small and not meaningful, while the cortical gains are substantial.
> > > >
> > > > > Jacobian regularity vs. Dice trade-off Ablations show increasing $\kappa$ improves Dice but degrades regularity. Do you have guidance on selecting $\kappa$ for different applications (e.g., longitudinal studies)? Could adaptive balancing be learned?
> > > >
> > > > For applications where diffeomorphic maps are required to perform quantitative analysis of shape change, we recommend using small values as $\kappa \in [1,2]$. These include applications such as statistical shape analysis, growth modeling, and patient-specific longitudinal studies. These settings maintain near-zero folding while preserving strong cortical and subcortical Dice. For cross-sectional or atlas-build tasks where segmentation accuracy is the primary goal, larger values ($\kappa \in [5,10]$) can be used to favor structural alignment at the cost of slightly reduced regularity. We do not recommend very large values of $\kappa$ (e.g. $\kappa=50$), as field regularity drops rapidly with negligible or no improvement in structural alignment. For $\kappa \leq 10$, we observe at worst $0.081\%$ folded voxels in our ablation on Table A.1. In practice, all these settings are acceptable for registration.
> > > >
> > > > Adaptive balancing is an interesting idea, thank you for suggesting that. One direction could be to use a curriculum learning schedule, where $\kappa$ is decreased over time, so the network can first learn to align structure then smooth registration later. A second, promising approach would be to use hypernetworks, which have demonstrated success in registration for selecting hyperparameters to balance structural alignment with field regularity (see HyperMorph, Hoopes et al., 2021). Such an approach can be beneficial for our work as we have several hyperparameters to select.
> > > >
> > > > We added the following text to Section 4.3 Ablations:
> > > >
> > > > *Hypernetworks can be considered to select hyperparameters at test-time (Hoopes et al., 2021)*

---

> > > > > ### Author Response · Authors · 2025-11-27
> > > > >
> > > > > >Topology enforcement While spherical alignment preserves cortical topology, volumetric deformation can still introduce folding. Would incorporating diffeomorphic constraints on the spherical path reduce the remaining ~0.1–0.2% folds?
> > > > >
> > > > > Both the spherical and volumetric networks already use stationary velocity fields (SVF) with scaling-and-squaring, so the transformations are diffeomorphic in the continuous model, in theory. The practical implementations inevitably violate some of the continuous assumptions: the velocity field is sampled on a finite grid, interpolation introduces discretization error, and the number of squaring steps limits the accuracy of the integration. Due to GPU memory constraints, we also predict an SVF at half-resolution in the volume registration network, then upsample by a factor of $2$ to get the full resolution. Increasing the alignment weight $\kappa$ also produces sharper local gradients which can break these assumptions and increase folding. Using more integration steps and a higher-resolution velocity field would reduce, but not completely eliminate, these folds.
> > > > >
> > > > > Even with these practical sources of numerical folding, the remaining ~0.1–0.2% folds are extremely small. Our method also produces substantially fewer folds than uGradICON (up to 0.96%) and CVS (up to 2.16%), indicating that the overall volumetric regularity is strong. We achieve significantly and substantially improved cortical correspondence with nearly diffeomorphic behavior.
> > > > >
> > > > > > Sphere parameterization distortion Stereographic projection introduces sampling and area distortion. How does this influence fine-scale folding alignment?
> > > > >
> > > > > Thank you for raising this point. You are correct that stereographic projection introduces sampling and area distortion. We address this in the following ways:
> > > > >
> > > > > - The spherical parameterization introduces denser sampling for higher latitude regions, i.e. the north and south poles are stretched the width of the image. To account for this, we perform distortion correction in each loss function by weighing samples by $\sin(\theta)$, where $\theta$ is the polar angle describing the elevation (Cheng et al., 2020; Li et al., 2024).
> > > > >
> > > > > - The parameterization introduces discontinuities across all borders of the image. We use a padding strategy from (Li et al., 2024). Specifically, we use a circular padding on the left and right borders, to replicate the “wrap around” effect when moving along the image. Along the top and bottom of the image (i.e. the north and south pole), we use a 180 degree circular shift followed by a reflection padding.
> > > > >
> > > > >  We have added text to Section 3.3 Discrete model to clarify:
> > > > >
> > > > > *To account for the nonuniform sampling of the spherical parameterization, where regions near the poles are stretched, we apply distortion correction in all surface-based losses by weighting samples by $\sin(\theta)$ (Cheng et al., 2020; Li et al., 2024), where $\theta$ is the elevation angle. We also handle the discontinuities at the image boundaries using the padding strategy of (Li et al., 2024), employing circular padding along the left–right axis and a $180^\circ$ circular shift with reflection padding at the poles.*
> > > > >
> > > > > > …Any alternatives considered (e.g., icosahedral CNNs)?
> > > > >
> > > > > We did consider icosahedral CNNs and spherical mesh convolutions, which provide more uniform sampling. However, these approaches require a rigid icosahedral tessellation that is not produced natively by standard neuroimaging tools (e.g., FreeSurfer outputs arbitrary-mesh spherical surfaces with subject-specific vertex ordering). Converting each subject into a consistent icosahedral mesh adds a nontrivial preprocessing pipeline and requires resampling functional or anatomical data onto a new mesh representation, which reduces compatibility with existing neuroscientific workflows. A key goal of our method is ease of adoption, to allow researchers to train directly on standard FreeSurfer outputs, we opted for a simple 2D parameterization. Our results show that this design choice still retains fine-scale folding alignment while substantially improving cortical correspondence.
> > > > >
> > > > > We have added the following text to Section 2 Related Work subsection Spherical Registration:
> > > > >
> > > > > *Icosahedral CNNs (Zhao et al., 2021) learn directly on the sphere but require a rigid tessellation that is not produced natively by standard neuroimaging pipelines.*

---

> > > > > > ### Author Response · Authors · 2025-11-27
> > > > > >
> > > > > > > Training pairs Registration direction is arbitrary. Did you augment by swapping moving/fixed pairs? Any asymmetry observed in deformation field quality?
> > > > > >
> > > > > > In training, at each iteration, we randomly sampled a target pair for each image. In expectation, this would lead to sampling pairs symmetrically. As we use a stationary velocity field parameterization of our network, symmetry is less of an issue as our deformation fields are diffeomorphic (in theory). We did not observe any asymmetry in deformation field quality.
> > > > > >
> > > > > > Runtime comparison fairness Since CVS requires spherical meshes at inference, did timing exclude preprocessing? Could UCS runtime remain competitive if including preprocessing of spheres during training on real pipelines?
> > > > > >
> > > > > > We did not include preprocessing time when reporting CVS results. CVS takes several hours after preprocessing to construct the cortical surface meshes and their inflated spheres.
> > > > > >
> > > > > > We have now clarified this in Section 4.2 Results:
> > > > > >
> > > > > > *In terms of computational time, CVS was slower than all methods, taking $2.5\pm0.5$ hours of optimization to register a pair of images.*
> > > > > >
> > > > > > > Scalability to groupwise registration Could your framework support joint template estimation or atlas building? What changes would be required to extend UCS to multi-subject alignment?
> > > > > >
> > > > > > This is an excellent idea, thank you for suggesting it. We are currently exploring ways to perform joint cortical-volumetric atlas construction. A possible method would be to extend current template construction methods that perform many iterative pairwise registrations using our trained model. Alternative formulations could combine existing spherical template construction frameworks with Atlas building frameworks (Dalca et al., 2019; Abulnaga et al., 2025, Ding and Niethammer, 2022). A challenge is in constructing the spherical template.
> > > > > >
> > > > > > We thank the reviewer kindly for the detailed review and many questions and feedback that has improved the paper.

---

### Official Review · Reviewer_rb9L · 2025-10-31

**Soundness:** 3
**Presentation:** 3
**Contribution:** 2
**Rating:** 4
**Confidence:** 4

**Summary:**

The paper proposes a deep learning framework called UCS that aligns both cortical and subcortical structures in in-vivo neuroimaging by using volume based dense registration combined with surface-based registration of the cortical surface. The paper posits the limitations of volumetric registration methods for registering the cortex despite its success in registering the subcortical structures and global anatomy. The problem with incorporating intensity and surface based registration is that the 2D surface registration has a gradient with a Lebesgue measure of 0 on the ambient 3D space, and to avoid this problem the paper proposes training a 2D and 3D registration network for the surface and volume respectively, and add a consistency term between the losses. Results are shown on four clinical MRI datasets (OASIS, ADNI, IXI, Mindboggle).

**Strengths:**

1. The mathematical formulation is very well defined, easy to follow, and technically sound. Usual papers in this topic are rather sloppy with overloaded notation or hand wavy definitions, but this paper is much clearer in that sense.
2. Most of the formulation is consistent with a typical deep learning registration framework, and the coupling term is the major technical novelty in the paper, which is easy to undeerstand.
3. Results are shown on a few popular clinical MRI datasets

**Weaknesses:**

1. The premise of the paper "While effective for aligning subcortical structures and global anatomy, volumetric deformable registration often fails in the cortex. The cortex is a thin, highly curved surface with significant inter-subject variability in folding patterns that is difficult to align in Euclidean space" is not substantiated in any way except for the experiments in the paper. Most methods trained on SynthSeg or Freesurfer labels (e.g. OASIS dataset) have achieved overall dice scores in the order of 0.88, almost to the point of overfitting on the labelmaps. Two of the labelmaps in freesurfer / synthseg labels are the cerebral cortex for each hemisphere. It would greatly strengthen the premise of the paper to show the performance of pre-trained optimization and deep learning methods on these datasets or labelmaps on only the cortical label registration.
2. Exaggerated narrative about slowness of optimization methods - "Moreover, because of its reliance on classical optimization techniques, CVS is computationally expensive, requiring several hours per subject pair, making it impractical for large-scale datasets." and other statements imply classical methods being very slow which has been shown to be false with newer implementations - particularly in the Oncoreg Learn2Reg challenge (where Syndeeds and ConvexAdam perform very well with no or little learning), and FireANTs which shows real-time registration on various clinical datasets.
3. Equation 2 is not a coupled optimization - coupling is only achieved with Equation 3. This is problematic because the objective (integral) in Equation 3 is only defined on the boundary of the surface $dS$ , which has a Lebesgue measure of 0 on the 3D ambient space. This suffers from the same problem as joint optimization of the volumetric and surface registration using a single volumetric displacement grid. The paper does not justify why Equation 3 does not suffer from the same problem that direct optimization on the surface matching objective using a volumetric displacement does. This is a major issue in the paper - since the whole idea behind not performing direct optimization of the cortical surface was the measure 0 gradient of the surface loss. Since this is the only proposed technical novelty in the paper (coupling the surface and volume registration), proper empirical (showing the gradient propagation with and without the coupling term) or theoretical (proofs) justification is needed.
4. "All CVS experiments are done on an Intel(R) Xeon(R) Gold 5218 CPU. (There is no GPU-based implementation of CVS.)" - Use FireANTs or ConvexAdam for GPUs. These baselines have existed for more than a year now, and are very efficient for the scale of clinical datasets.
5. Baselines are not adequate. Voxelmorph with segmentation loss should be compared with UCS. Synthmorph and unigradICON are general purpose registration tools like ANTs, but since the method assumes access to a training dataset, at least a few baselines should be trained with the same data.
6. Line 424: "Including our proposed sphere loss (Base+Dice(all)+Sphere) produces a clear increase in Dice (cortical) while preserving a high Dice(subcortical) score." - this is not the right ablation to run. The correct ablation here is to add the cortical surface loss directly in the training / inference objective instead of the roundabout way (i.e. consistency between network outputs).

Minor issues:
1. The geometric descriptors used in the surface registration are not mentioned. This hinders reproducibility of the paper.
2. to my knowledge, loss masking in unigradicon does not explicitly align the structures themselves but only registers within the provided masks. im not sure why this baseline is used in that case
3. Line 469 - "Unlike other methods that require additional inputs (cortical meshes, segmentations) at inference, our method requires only structural MRI images."  - This is a partially false statement - since all methods considered in the paper (except unigradICON with masking) do not require additional labels at inference. This is only true for classical iterative methods that do not learn any label-aware features.
4. Another potential weakness in the evaluation setup is that the considered datasets are possibly overfit to in the long run. There are other high resolution, high quality datasets (e.g. Ultracortex 9.4T) that have higher than 1mm resolution (i.e. 0.6mm)  and manual labellings of the cortical surface - providing a challenging evaluation setup for the exact problem tackled in the paper

**Questions:**

1. 2D registration has to be performed on a sphere where the coordinate $\phi$ wraps around from 0 to $2\pi$. how is this circular nature considered in the spherical registration? appropriate citations or derivations in the supplementary material can be useful
2. What is the performance of the method without the segmentation loss? This might be used for hippocampus or MTL registration for example where only the registration of a single surface might be of interest without registering other structures.
3. In many other clinical or research scenarios pertinent to neuroimaging studies, the surface mesh is typically available (using a segmentation algorithm or manually) and accurate registration of the surface and volume might be desired. However, collecting training data across many subjects might be infeasible. In such a scenario, can UCS be used in an online optimization fashion?

I would like the authors to address some of the weaknesses of the paper followed up by the questions in this section.

---

> ### Author Response · Authors · 2025-11-26
>
> We thank the reviewer for commenting on the strength and technical soundedness of the mathematical notation, and clarity in notation and exposition. We appreciate the acknowledgment of the major technical novelty in the paper and strength of experimental results.
>
> > The premise of the paper "While effective for aligning subcortical structures and global anatomy, volumetric deformable registration often fails in the cortex. ... It would greatly strengthen the premise of the paper to show the performance of pre-trained optimization and deep learning methods on these datasets or labelmaps on only the cortical label registration.
>
> Thank you for raising this point. We have included results on the cerebral cortex for each hemisphere, which we label as Dice (CC), CC indicating “cerebral cortex”. We include the results in **new Supplemental Section A.2 Whole Cerebral Cortex Alignment Performance.**
>
> On these labelmaps, volumetric baselines show stronger performance than when considering the cortical parcellations. For example, uGradICON improves from 0.58 Dice on cortical parcellations to 0.71 Dice on whole cerebral cortex segmentation in OASIS-1 and ADNI. Evaluating alignment on the entire cerebral cortex highlights the strength of our approach and the importance of spherical registration. All volumetric-only baselines align the entire cerebral cortex volume reasonably well but perform substantially worse on cortical parcellations. This gap reflects the need for geometric features to align homologous cortical regions (Fischl et al., 1999). In contrast, our method achieves consistently high Dice on both cerebral cortex and cortical labels across datasets, demonstrating its effectiveness and utility for structural and functional neuroimaging studies.
>
> > Exaggerated narrative about slowness of optimization methods - "Moreover, because of its reliance on classical optimization techniques, CVS is computationally expensive, requiring several hours per subject pair, making it impractical for large-scale datasets."... newer implementations - particularly in the Oncoreg Learn2Reg challenge (where Syndeeds and ConvexAdam perform very well with no or little learning), and FireANTs which shows real-time registration on various clinical datasets.
>
> Thank you for pointing this out. We agree that there are modern optimization-based methods such as FireANTs that show rapid convergence. Our point was intended to refer to CVS specifically, which is not optimized for computational efficiency nor has a GPU implementation. As CVS is the state-of-the-art method for volumetric registration of the cortex and subcortex, this was our main point of comparison. We have clarified this text in Section 2 Related Work:
>
> *A strength of these variational approaches is that they offer a principled way to propagate surface constraints into the volume using well-understood physical models. However, they remain impractical for large-scale datasets, as they require several hours of computation per pair and depend on extracting cortical surfaces and segmentation labelmaps.*
>
> > All CVS experiments are done on an Intel(R) Xeon(R) Gold 5218 CPU. (There is no GPU-based implementation of CVS.)" - Use FireANTs or ConvexAdam for GPUs. These baselines have existed for more than a year now, and are very efficient for the scale of clinical datasets.
>
> Per your request, we have added FireANTs as an additional baseline and report performance on Tables 1 and A.2 and Figure 2. We also report that FireANTs takes only seconds of optimization in Sections 4.2 Results and 4.1 Experimental Setup, subsection Baselines:
>
> *We also evaluate with using the SyGN algorithm in FireANTs (Jena et al., 2025). FireANTs is a fast, GPU-accelerated, multi-scale
> diffeomorphic registration framework that performs pairwise optimzation.*
>
> The inclusion of FireANTs does not change the results of our paper, as our method outperforms across both structural alignment and map bijectivity metrics.
>
> > Baselines are not adequate. Voxelmorph with segmentation loss should be compared with UCS. ...
> since the method assumes access to a training dataset, at least a few baselines should be trained with the same data.
>
> Thank you for noting that we should include a baseline trained with the same data. We have added VoxelMorph and trained it on our same training set, with the same neural network architecture and number of parameters. We have also added FireANTs as a baseline. We note that both FireANTs and CVS are optimized on all datasets.
>
> We include the results of VoxelMorph and FireANTs on Table 1, Table A.2, Figure 2, and Section 4.2 Results. As expected, VoxelMorph performs poorly in Dice (cortical), achieving similar performance to other volumetric baseline methods. However, VoxelMorph achieves consistently strong Dice (subcortical) performance.
>
> We also clarify that the IXI and Mindboggle datasets, used in our experiments, were held out from the training of UCS, and represent dataset generalization.

---

> > ### Author Response · Authors · 2025-11-26
> >
> > > Equation 2 is not a coupled optimization - coupling is only achieved with Equation 3. This is problematic because the objective (integral) in Equation 3 is only defined on the boundary of the surface  , which has a Lebesgue measure of 0 on the 3D ambient space. This suffers from the same problem as joint optimization of the volumetric and surface registration using a single volumetric displacement grid. The paper does not justify why Equation 3 does not suffer from the same problem that direct optimization on the surface matching objective using a volumetric displacement does. This is a major issue in the paper - since the whole idea behind not performing direct optimization of the cortical surface was the measure 0 gradient of the surface loss. Since this is the only proposed technical novelty in the paper (coupling the surface and volume registration), proper empirical (showing the gradient propagation with and without the coupling term) or theoretical (proofs) justification is needed.
> >
> > We thank the reviewer for this question. We did originally mention that the coupling energy in Eq (3) operating on the surface $\partial M$ is a set of Lebesgue measure zero in $\mathbb{R}^3$. We agree with you that in the continuous formulation this can suffer from decoupling to the volumetric deformation.
> > However we highlight that our **discrete implementation** does not suffer from these issues:
> > - In our implementation the consistency loss evaluates the volumetric field $\varphi$ at mesh vertex locations, using differentiable trilinear interpolation. The error signal at a zero-measure vertex (or its associated surface voronoi area) is therefore distributed to the eight neighboring voxels in the grid. Surface gradients thus directly update the deformation field.
> > - The volumetric deformation is not a set of independent vectors, but the output of a UNet. This imposes a strong smoothness prior. Local constraints can therefore have a wide influence, as the network learns cross-correlation between voxels.
> > - Finally, we use an explicit diffeomorphic prior on the volumetric deformation, which (thanks to the discretization) forces the surface deformation to propagate to the volume.
> >
> > We address this point in Section 3.2 Spherical alignment:
> >
> > *Note that this constraint only acts on a set of Lebesgue measure zero in $\mathbb{R}^3$ (the cortical surface $\partial M_1$) and is therefore weak. In practice, however, our discrete implementation avoids this issue: the consistency loss samples the volumetric deformation at mesh vertex locations via trilinear interpolation, distributing gradients to neighboring voxels. We also impose smoothness priors on the map, allowing the surface-derived updates to naturally propagate throughout the volume.*
> >
> > > Line 424: "Including our proposed sphere loss (Base+Dice(all)+Sphere) produces a clear increase in Dice (cortical) while preserving a high Dice(subcortical) score." - this is not the right ablation to run. The correct ablation here is to add the cortical surface loss directly in the training / inference objective instead of the roundabout way (i.e. consistency between network outputs).
> >
> > The fixed and moving cortical surface meshes have different connectivity, and so formulating a loss to align them without the consistency term is not straightforward. If the reviewer has specific loss functions in mind to include, we would be happy to do so. The advantage of the consistency term is that it is agnostic to mesh connectivity, and uses the spherical alignment as a proxy supervision signal. Finally, the corpus of cortical surface alignment literature demonstrates that using a spherical domain is necessary to maintain topological consistency when aligning cortical surfaces that have large inter-subject variability.
> >
> > > The geometric descriptors used in the surface registration are not mentioned. This hinders reproducibility of the paper.
> >
> > We did mention in Section 3.3.1 Unified Training and Cortical Alignment, paragraph “Image Similarity Loss” that we precompute the sulcal depth and mean curvature as our geometric features. These are the standard features used in the FreeSurfer pipeline, and the features used by CVS.
> >
> > > to my knowledge, loss masking in unigradicon does not explicitly align the structures themselves but only registers within the provided masks. im not sure why this baseline is used in that case.
> >
> > We agree. We now clarify that we included this baseline as the provided masks are likely to improve the test-time optimization of uGradICON to focus on the cortical area. Experimentally, we see a consistent increase in Dice (cortical) with this baseline over the standard uGradICON.
> >
> > In Section 4.1 Experimental Setup, paragraph “Baselines.” we write:
> >
> > *We also use the uGradICON-seg variant that performs test-time optimization using segmentation labelmaps to improve cortical structure alignment.*

---

> > > ### Author Response · Authors · 2025-11-26
> > >
> > > > Line 469 - "Unlike other methods that require additional inputs (cortical meshes, segmentations) at inference, our method requires only structural MRI images." - This is a partially false statement - since all methods considered in the paper (except unigradICON with masking) do not require additional labels at inference. This is only true for classical iterative methods that do not learn any label-aware features.
> > >
> > > Thank you for pointing this out. You are correct, we intended this sentence to refer to classical iterative methods, and CVS specifically. We have corrected the text to say:
> > >
> > > *Unlike the state-of-the-art iterative method for combined volume and surface registration (Postelnicu
> > > et al., 2008), we do not require additional inputs (cortical meshes, segmentations) at inference. Our method only requires structural MRI images.*
> > >
> > > > Another potential weakness in the evaluation setup is that the considered datasets are possibly overfit to in the long run. There are other high resolution, high quality datasets (e.g. Ultracortex 9.4T) that have higher than 1mm resolution (i.e. 0.6mm) and manual labellings of the cortical surface - providing a challenging evaluation setup for the exact problem tackled in the paper
> > >
> > > We are working on implementing this suggested experiment and will respond in 1-2 days with these results. We would like to note that this dataset does not have manual cortical parcellations, only gross segmentations for the gray matter.
> > >
> > > ## Questions
> > >
> > > >2D registration has to be performed on a sphere where the coordinate
> > >  wraps around from 0 to 2$\pi$. how is this circular nature considered in the spherical registration? appropriate citations or derivations in the supplementary material can be useful
> > >
> > > You are correct that there are several aspects to be considered when using a spherical parameterization. We do the following:
> > >
> > > - The spherical parameterization introduces denser sampling for higher latitude regions, i.e. the north and south poles are stretched the width of the image. To account for this, we perform distortion correction in each loss function by weighing samples by $\sin(\theta)$, where $\theta$ is the polar angle describing the elevation (Cheng et al., 2020; Li et al., 2024).
> > > - The parameterization introduces discontinuities across all borders of the image. We use a padding strategy from (Li et al., 2024). Specifically, we use a circular padding on the left and right borders, to replicate the “wrap around” effect when moving along the image. Along the top and bottom of the image (i.e. the north and south pole), we use a 180 degree circular shift followed by a reflection padding.
> > >
> > > We have added text to Section 3.3 Discrete model to clarify:
> > >
> > > *To account for the nonuniform sampling of the spherical parameterization, where regions near the poles are stretched, we apply distortion correction in all surface-based losses by weighting samples by $\sin(\theta)$ (Cheng et al., 2020; Li et al., 2024), where $\theta$ is the elevation angle. We also handle the discontinuities at the image boundaries using the padding strategy of (Li et al., 2024), employing circular padding along the left–right axis and a $180^\circ$ circular shift with reflection padding at the poles.*
> > >
> > > > What is the performance of the method without the segmentation loss? This might be used for hippocampus or MTL registration for example where only the registration of a single surface might be of interest without registering other structures.
> > >
> > > As demonstrated in our ablations on Table 2, the performance without the segmentation loss is slightly better on Dice (cortical) than with, when comparing `Base+Sphere` against `Base+Dice(all)`.
> > >
> > > You raise a good point that this method could be useful for registration of other genus-0 structures, such as the hippocampus. Extending our method to also register these small structures is likely to provide additional supervision to improve alignment.
> > >
> > > We have added text to the Limitations and Future Work section:
> > >
> > > *A benefit of our formulation is that it can be easily extended to include additional structures with genus-0 topology. For example, this can improve alignment of the hippocampus by providing additional supervision through surface alignment.*

---

> > > > ### Author Response · Authors · 2025-11-26
> > > >
> > > > > In many other clinical or research scenarios pertinent to neuroimaging studies, the surface mesh is typically available (using a segmentation algorithm or manually) and accurate registration of the surface and volume might be desired. However, collecting training data across many subjects might be infeasible. In such a scenario, can UCS be used in an online optimization fashion?
> > > >
> > > > This is an excellent point. In such scenarios, UCS can indeed be used in an online or test-time optimization, as you mentioned. This would involve fine-tuning our pretrained volumetric and surface networks using the consistency loss together with their respective registration losses.
> > > >
> > > > The only additional requirement is obtaining the spherical parameterization of the cortical surface at test time. This can be done in several standard ways, for example using the FreeSurfer’s `mris_inflate` function or using the HCP/Connectome Workbench, or any other widely used cortical inflation tool. Modern, deep learning based approaches such as TopoFit (Hoopes et al., PMLR 2022) and Vox2Cortex (Bongratz et al., CVPR 2022) can also be easily integrated into an online pipeline. UCS is agnostic to the specific inflation algorithm, we only require a one-to-one correspondence between the cortical surface mesh and inflated sphere.

---

### Official Review · Reviewer_7rxY · 2025-10-31

**Soundness:** 3
**Presentation:** 3
**Contribution:** 1
**Rating:** 2
**Confidence:** 5

**Summary:**

This paper presents UCS, a deep learning framework for brain MRI registration that combines volumetric and surface-based approaches. UCS employs a coupled architecture with spherical coordinate mappings and a consistency loss to enforce geometric coherence, using surface mesh registration results as additional supervision for volumetric registration. The method achieves state-of-the-art performance across multiple datasets.

**Strengths:**

1. UCS couples volumetric and spherical registration networks, promoting anatomical consistency across domains.

2. UCS outperforms classical (CVS) and modern deep learning methods (SynthMorph, uniGradICON) in both cortical and subcortical Dice scores.

**Weaknesses:**

1. The authors claim that their approach provides a unified solution for the registration of cortical and subcortical structures. And the key idea is to use correspondence between surface areas as additional supervision signal. This idea is not novel; a similar approach was presented in https://doi.org/10.1016/j.media.2019.101540

2. From the title, I had the expectation that the proposed approach would have some special designs for the registration of subcortical structures. However, such designs were not found.

**Questions:**

I don't have questions to ask because the paper is clearly written and the approach is straightforward to understand.

---

> ### Author Response · Authors · 2025-11-26
>
> We thank the reviewer for acknowledging the strength in our experimental results, methodological formulation, and clarity of exposition.
>
> > The authors claim that their approach provides a unified solution for the registration of cortical and subcortical structures. And the key idea is to use correspondence between surface areas as additional supervision signal. This idea is not novel; a similar approach was presented in https://doi.org/10.1016/j.media.2019.101540
>
> Thank you for highlighting this connection. The referenced work (Ahmad et al., 2019) is an important demonstration of how cortical surface information can improve volumetric registration, particularly in the context of infant brain imaging. Their iterative method was designed specifically for early developmental brains, where cortical folding patterns, tissue contrasts, and anatomical variability differ substantially from adult brain MRI.
>
> Our approach is inspired by this line of work, through CVS (Postelnicu et al., 2008), which is our primary baseline. We agree that supervising volumetric registration by surface correspondence is a powerful idea that had been previously proposed. Our goal is to model this principle in a modern learning-based, diffeomorphic framework. Our work targets a different setting, by building a general-purpose registration model for adult datasets, which can be integrated with large neuroimaging pipelines.
>
> Ahmad et al. 2019 incorporate surface correspondence into the volumetric deformation through a variational optimization with an elastic regularizer and surface force terms. In our case, both the spherical branch and the volumetric branch are neural network models that output stationary velocity fields. Our models are trained end-to-end, and coupled through our proposed differentiable consistency loss that directly enforces geometric agreement between spherical and 3D representations. This enables the cortical signal to propagate into subcortical alignment through a single learned diffeomorphic field, while retaining compatibility with contemporary neural registration pipelines. Further, our consistency formulation is simple and adaptable to other types of loss functions for volumetric-surface consistency.
>
> An additional strength of our method is that unlike prior work (Postelnicu et al., 2008; Joshi et al. 2009; Ahmad et al. 2019), we do not require cortical surface meshes or segmentation maps at inference (optimization) time. We can predict diffeomorphic registration fields in milliseconds, without the need for complicated preprocessing pipelines. This allows our method to be fully integrated into modern neuroimage analysis pipelines. Finally, we achieve substantially better performance in cortical alignment than several baseline optimization and deep learning-based methods while maintaining regular deformation fields with few folds.
> We also contacted the authors of Ahmed et al. (2019) for an implementation to perform a direct comparison, but we were unable to obtain a copy of the code.
>
> We have revised the manuscript to clarify this relationship and ensure appropriate acknowledgment. Specifically, we modify Section 2 Related Work:
>
> **Combined Volume and Surface Registration.** *Existing neuroscientific studies treat volumetric and cortical registration as two separate problems, often combining their outputs through {\it ad hoc} procedures. The Combined Volume and Surface framework (CVS) (Postelnicu et al., 2008; Zöllei et al., 2010) is the standard method for jointly aligning cortical and subcortical anatomy in adult brains. CVS first aligns the cortical surface then propagates the resulting deformation into the volume using a nonlinear elastic model, followed by intensity-based refinement. While effective, this sequential formulation decouples the surface and volumetric objectives, can introduce inconsistencies near the cortical–subcortical boundary, and requires surface meshes and long runtimes. Related work has also used cortical correspondence to supervise volumetric deformation in variational settings (Ahmad et al., 2019), though focused in the infant-brain domain. A strength of these variational approaches is that they offer a principled way to propagate surface constraints into the volume using well-understood physical models. However, they remain impractical for large-scale datasets, as they require several hours of computation per pair and depend on extracting cortical surfaces and segmentation labelmaps.*
>
> *To address these limitations, we introduce UCS, which incorporates the established idea of surface-guided alignment into a single, unified learning-based diffeomorphic framework. UCS jointly aligns cortical and subcortical structures through a single forward pass, requires only 3D MRI at inference time (without volumetric segmentations and surface meshes), and provides the speed and simplicity needed for high-throughput neuroimaging studies.*

---

> > ### Author Response · Authors · 2025-11-26
> >
> > > From the title, I had the expectation that the proposed approach would have some special designs for the registration of subcortical structures. However, such designs were not found.
> >
> > Our goal in this work was to develop a straightforward, unified formulation that improves cortical alignment while remaining fully compatible with existing volumetric registration frameworks. Rather than adding a separate subcortical module, our approach couples the spherical cortical alignment and the volumetric alignment through a single smooth diffeomorphic field. This joint field necessarily propagates improved cortical correspondence into the underlying subcortical structures, which is why our method still performs competitively on subcortical Dice without any subcortical-specific design.
> >
> > We intentionally did not introduce additional subcortical mechanisms because a key strength of our approach is that it can function as a plug-in component for standard volumetric models: it requires no new architectural branches, no dataset-specific subcortical labels, and no extra preprocessing beyond standard FreeSurfer surfaces for training. The method’s simplicity enables drop-in use with commonly adopted neuroimaging pipelines while still yielding substantial improvements in cortical alignment and performing as well as, or better than, state-of-the art methods in subcortical structural alignment, which is why they are included in the title.

---

### Official Review · Reviewer_57Lz · 2025-11-01

**Soundness:** 3
**Presentation:** 3
**Contribution:** 3
**Rating:** 4
**Confidence:** 3

**Summary:**

This paper proposes UCS, a unified learning framework that trains a 3D diffeomorphic registration network and a 2D spherical registration network jointly, with a cortical-consistency loss that ties the cortical ribbon deformation in the volume to the sphere-space deformation.
The writing is clear and the experiments are sufficient. But still, some weakness exists.

I am inclined to give a score of around 5, but since that option isn't available, I will select 4 for now.

**Strengths:**

- This paper proposes a real and valuable challenge in MRI registration: current pipelines split surface vs volume.
- Key technical idea (soft consistency energy on cortical surface) is simple, plausible, and seems to explain gains; ablation supports this.
- Sufficient comparison to the right classical joint method (CVS) and to current learning baselines (SynthMorph variants, uniGradICON).
- It is great to test on two additional held-out datasets.

**Weaknesses:**

- The author claims, "Unlike other methods that require additional inputs (cortical meshes, segmentations) at inference, our method requires only structural MRI images." However, they also mention, "For each image, we use FreeSurfer to generate anatomical segmentations" in the experiment setup. Therefore, it is not truly "only structural MRI images" as input. If you plan to integrate this with your model as a complete pipeline, do you include the time spent on segmentation when comparing to CVS?
- Why this task cannot be done by “3D reg + surface reg, then post-hoc propagation” process? Compared to some baseline methods, they did missing one modality in registration. It is quite nature to think more modality for cross-validation is better. The reviewer would like to see a discussion or clarification on this part

**Questions:**

- Discuss licensing/availability of FreeSurfer usage. Many ICLR readers are not neuroimaging specialists.

---

> ### Author Response · Authors · 2025-11-26
>
> We thank the reviewer for the kind words about this work and recognizing the experimental strengths of our approach, sound formulation, and that our work addresses a real and valuable challenge in MRI registration.
>
> >The author claims, "Unlike other methods that require additional inputs (cortical meshes, segmentations) at inference, our method requires only structural MRI images." ... If you plan to integrate this with your model as a complete pipeline, do you include the time spent on segmentation when comparing to CVS?
>
> Thank you for bringing up this point. We would like to clarify that our method does indeed use the FreeSurfer `recon_all` pipeline to prepare data for **training**. In training, we use the MRI volumes, the cortical and subcortical segmentations, and the cortical surface and spherical meshes.
>
> However, the most important distinction is that in **inference**, after having trained the model, we do not require the segmentations nor the meshes. We only require the T1w MRI image. Our only preprocessing step is affine alignment to a template image, which takes on the order of seconds, and we perform this same preprocessing for uGradICON and SynthMorph.
>
> To clarify the runtime, the model predicts the 3D deformation field that jointly aligns the subcortical and cortical structures in a single forward pass, taking milliseconds on a GPU, or seconds on a CPU. In contrast, CVS requires segmentation, meshes, and cortical surfaces for each optimization, taking approximately 2.5 hours, plus the time to run the FreeSurfer recon_all pipeline, which is another 2-6 hours.
>
> To clarify, this point, we have modified the text in Section 4.1 Experimental Setup:
>
> **Processing**. *We use FreeSurfer (Fischl, 2012) to perform all image preprocessing. FreeSurfer is an open-source, widely used toolkit in neuroimaging that supports cortical surface analysis, longitudinal workflows, and population-scale MRI studies. We first perform affine alignment of each image to the Talairach template (Talairach & Tournoux, 1990).
> We crop all images to a roughly 10 mm margin around the brain matter. To prepare our training data, we generate anatomical segmentations, internal and external cortical surface reconstructions, and spherically inflated surfaces with parameterized curvature maps. For evaluation, we use both volumetric segmentations of subcortical brain regions, computed on the volume, as well as parcellations of the cortical ribbon, computed on the surface mesh. Surface parcellations are rasterized as volumetric segmentation. We emphasize that our method does not require surface meshes, spheres, nor segmentation labelmaps at inference.*
>
> > Discuss licensing/availability of FreeSurfer usage. Many ICLR readers are not neuroimaging specialists.
>
> We have added the following text to Section 4.1 Experimental Setup:
>
> *We use FreeSurfer (Fischl, 2012) to perform all image preprocessing. FreeSurfer is an open-source, widely used toolkit in neuroimaging that supports cortical surface analysis, longitudinal workflows, and population-scale MRI studies.*

---

> > ### Author Response · Authors · 2025-11-26
> >
> > > Why this task cannot be done by “3D reg + surface reg, then post-hoc propagation” process?
> >
> > Thank you for bringing up this important point. The optimization problem is not well-defined because the two registrations produce deformation fields on two different domains. The 3D registration yields a volumetric warp defined throughout the brain, while the surface registration produces a warp defined only on the cortical manifold, represented as a spherical displacement field. There is no uniquely determined extension of a surface warp to the interior volume, as this is an underdetermined problem that requires additional regularization. Extending the surface displacement requires imposing a physical or geometric regularization, for example solving a Laplacian or elastic PDE where the surface map is used as a boundary condition.
> >
> > In practice, the volumetric registration may converge to a local minima that is incompatible with the desired surface alignment. A post-hoc correction is unlikely to undo such a minima without substantially altering the volumetric field, and the propagated deformation is not guaranteed to remain diffeomorphic or consistent with the original 3D map. As a result, the sequential strategy generally cannot ensure a single coherent deformation that simultaneously satisfies both cortical and subcortical alignment objectives.
> >
> > Several prior works already recognize this issue and explicitly formulate joint surface-volume registration to avoid inconsistencies that arise when the surface and volumetric maps are computed separately (Joshi et al., 2009; Postelnicu et al., 2008; Ahmad et al., 2019). We compare against the state-of-the-art method for adult registration, CVS, and our experiments demonstrate this approach produces many folded voxels (up to 2%) and substantially weaker cortical and subcortical alignment.
> >
> > To clarify, we modify Section 1 Introduction:
> >
> > *Using fundamentally different representations for surface- and volume-based registration forces neuroscience researchers to tackle two disjoint problems and integrate solutions *ad hoc* (Tucholka et al., 2012). The integration process involves solving an elastic partial differential equation to propagate the surface registration to the interior brain. However, this approach is not guaranteed to stay consistent with the original volumetric registration, preventing a single coherent registration that simultaneously satisfies both cortical and subcortical alignment objectives. This introduces errors, undermining whole-brain analyses by potentially misaligning anatomically and functionally connected areas (Joshi et al., 2009; Postelnicu et al., 2008; Ahmad et al., 2019).*
> >
> > ### References
> > Anand Joshi, Richard Leahy, Arthur W Toga, and David Shattuck. A framework for brain registration
> > via simultaneous surface and volume flow. In International Conference on Information Processing
> > in Medical Imaging, pp. 576–588. Springer, 2009
> >
> > Alan Tucholka, Virgile Fritsch, Jean-Baptiste Poline, and Bertrand Thirion. An empirical comparison
> > of surface-based and volume-based group studies in neuroimaging. Neuroimage, 63(3):1443–1453,
> > 2012.
> >
> > Sahar Ahmad, Zhengwang Wu, Gang Li, Li Wang, Weili Lin, Pew-Thian Yap, and Dinggang Shen.
> > Surface-constrained volumetric registration for the early developing brain. Medical Image Analysis,
> > 58:101540, 2019.
> >
> > Gheorghe Postelnicu, Lilla Zollei, and Bruce Fischl. Combined volumetric and surface registration.
> > IEEE transactions on medical imaging, 28(4):508–522, 2008.

---

> > > ### Author Response · Authors · 2025-11-26
> > >
> > > > Compared to some baseline methods, they did missing one modality in registration. It is quite nature to think more modality for cross-validation is better. The reviewer would like to see a discussion or clarification on this part
> > >
> > > We thank the reviewer for raising the question of evaluating our method on additional imaging modalities. This is an important point, and we agree that multimodal validation is a valuable direction for future work.
> > >
> > > T1w MRI is the established standard for cortical surface reconstruction because it provides the strongest gray-white matter contrast and supports reliable delineation of white and pial surfaces (Dale et al., 1999; Fischl, 2012). Other modalities such as T2w, FLAIR, or diffusion-weighted MRI provide complementary tissue information, but they do not offer sufficient contrast to independently extract cortical surfaces. As a result, high-quality cortical ground truth cortical surfaces are always extracted from T1w MRI, and T1w is the modality available in neuroscientific research studies that perform cortical surface-based analysis.
> > >
> > > Foundation model baselines (SynthMorph, uGradICON) can indeed compare against other modalities, as you mentioned. However, these baselines were not designed explicitly for cortical structure alignment, and perform poorly on this task as demonstrated on Table 1. In contrast, CVS (Postinelcu et al., 2008) is the state-of-the-art registration method for adult cortical and subcortical structural alignment. CVS **only operates on T1w images** for the reasons mentioned above. Thus, our evaluation using T1w images is consistent with standard neuroimaging pipelines for this task.
> > >
> > > However, you do raise an interesting point, and we plan to build on this direction in future work. In training, our model requires cortical surfaces corresponding to MRI images. We plan to construct a dataset with additional modalities such as T2w or FLAIR aligned to T1w images with cortical surface meshes. We can then train our model jointly on these imaging modalities. An advantage unique to our method is that we can then perform inference on these additional modalities **without requiring cortical surface meshes**. Unfortunately, the rebuttal period for ICLR did not allow sufficient time to explore this direction.
> > >
> > > To clarify this point, we made the following changes to the text:
> > >
> > > In Section 4.1 Experimental Setup, paragraph Held-out datasets:
> > >
> > > *All evaluations are performed on T1w MRI, which is the modality that enables reliable cortical surface extraction. T1w MRI has the strongest gray-white matter contrast and supports reliable delineation of white and pial surfaces (Fischl et al., 1999; Fischl, 2012; Hoopes et al., 2022).*
> > >
> > > Under Section 5, Limitations:
> > >
> > > *Our model was also only tested on the T1w modality. T1w is the most widely used imaging modality for structural imaging studies (Bhalerao et al., 2024), and it is currently the only modality from which cortical surfaces can be reliably reconstructed (Fischl 2012). An interesting future direction is extending unified cortical–subcortical registration to multimodal acquisitions. Additional contrasts such as T2w, FLAIR, or DWI could provide complementary information for subcortical tissue segmentation or characterization. Incorporating these modalities would require multimodal training with all contrasts aligned to the T1w image so that the cortical surfaces remain consistent across channels. Importantly, at inference time, the surfaces would no longer be required, potentially reducing the current dependence on T1w imaging for surface extraction.*

---

### Author Response · Authors · 2025-11-27
**Summary of Response to Reviewers**

We sincerely thank all reviewers for their detailed and insightful feedback, constructive suggestions, interest in our work, and time spent reviewing. The feedback has been incorporated to improve the paper. Revised text in the manuscript appears in red.

We are happy that the reviewers felt the paper addressed a real and valuable challenge in neuroimaging [`57Lz`,`rb9L`,`L2kZ`], that the formulation was technically sound, rigorous, and novel [`rb9L`,`L2kZ`], well written and easy to follow [`57Lz`,`7rxY`,`rb9L`,`L2kZ`], with substantially improved results across several datasets [`57Lz`,`7rxY`,`rb9L`,`L2kZ`].

To address specific concerns and incorporate feedback:

- Reviewer `57Lz`: We have improved the writing by distinguishing computational time differences between our method and CVS, formalized the need for our method which unifies spherical and volumetric registration rather than performing sequential or *post-hoc* alignment, and addressed future work incorporating additional modalities.

- Reviewer `7rxY`: We introduced textual changes to clarify how our work builds on earlier surface-guided alignment approaches such as Ahmad et al. (2019) and CVS (Postelnicu et al., 2008). We also clarified the specific contributions and novelties of our approach that differentiate it within this line of work, both in its mathematical formulation and in its practical design.

- Reviewer `rb9L`: We expanded the manuscript to address the central concerns by adding new experiments, clarifying theoretical points, and refining our comparisons to prior work. We now report whole cerebral cortex performance, include two additional baselines: VoxelMorph and FireANTs, and provide clearer justification for our coupling formulation in the discrete setting.

- Reviewer `L2kZ`: We expanded several sections of the manuscript to provide clearer justification for key design choices, including preprocessing requirements, spherical parameterization, and architectural decisions. We also strengthened the Discussion and Limitations to better contextualize the scope, assumptions, and robustness of our approach, and to highlight promising directions for future work.

All individual points are addressed in the individual responses below. We greatly appreciate the feedback and interest in this work, and we welcome any further discussion.

---

### Author Response · Authors · 2025-12-01
**Rebuttal Summary**

Dear Area Chair,

Please find a summary of the main concerns of all reviewers and our efforts to address them to make the paper stronger.

Reviewers `57Lz`,`rb9L`, and `L2kZ` had largely positive feedback, and their concerns were directly addressable and strengthened the paper. These reviewers agreed that the work provided a novel contribution to solve a real and challenging neuroimaging problem. Reviewers found the work technically sound and rigorous [`rb9L`,`L2kZ`] and all four reviewers acknowledged the substantially improved results over several baselines [`57Lz`,`rb9L`,`L2kZ`,`7rxY`] . Reviewer `7rxY`’s primary criticism was about the novelty of this work, which we carefully addressed in our response.

We summarize their main concerns and our main changes.

- For reviewer `57Lz`, we clarified, both in the response and in the paper revision, aspects about the preprocessing time, what inputs our method requires at inference, the need for our solution, and future work. We expanded the introduction to include mathematical justification for why a post-hoc heuristic approach is insufficient for this task. We also described in detail the preprocessing steps and clarified computational times and input data requirements. Lastly, we provided justification for focusing on T1w images and speculated future work to expand on additional modalities. In their original review summary, the reviewer indicated that they are inclined to give the paper a higher score given a satisfying author response, and we believe that we have achieved this.

- Reviewer `7rxY`’s primary concern was about the novelty of the proposed idea. They cited “Surface-constrained volumetric registration for the early developing brain” by Ahmad et al., MedIA 2019, claiming that this work introduced surface-supervision for volumetric registration. This claim is false, as CVS (Postelnicu et al., 2008) is the first to do so, which we discuss at length in the paper. Ahmad et al. (2019), is a small modification from CVS (Postelnicu et al., 2008) driven by the analysis of infant MRI instead of adult MRI. In the paper, we use CVS as the main baseline, and introduce several innovations and novelties beyond **just** using surface supervision in registration. To address this concern, we introduced several changes to the Related Work section distinguishing the proposed work from both CVS and Ahmad et al. We also reached out to the authors of Ahmad et al., to get the code, but they did not respond.

- Reviewer `rb9L` requested additional additional baselines, evaluation metrics, and clarifications about computational time and algorithmic choices. We addressed these by evaluating the two baselines that they requested (FireANTs and VoxelMorph) on all datasets. They also requested evaluation on the entire cerebral cortex labelmap, which we included. We continue to outperform the additional baselines on the 3 datasets used in evaluation. We also outperform all other methods on cerebral cortex structural alignment. Finally, we now clarify reported computational time, mathematical formulation, and implementation choices in the paper.

- Reviewer `L2kZ` requested additional discussion on method robustness, differences in experimental performance, and clarity in the math and figures. The reviewer also posed several interesting questions and ideas for future work. We addressed their concerns by significantly expanding the Limitations and Future Work section to discuss robustness of our method, clarified the main architectural figure and preprocessing steps, key design decisions, and expanded discussion on future work in expanding our method to other surface representations, clinical scans, and data with pathology. We also provided detailed answers to many of their questions which demonstrated strong interest in the proposed method. This reviewer was very positive about this work, as also demonstrated in their high initial score.

Given the substantial changes that included several expositional changes, expanded experimentation and baselines, we believe that we have satisfied the reviewers’ primary concerns and would have had a fruitful discussion period. Overall, the reviewers had positive sentiment about this work and their reviews were mostly constructive and enthusiastic.

---

### Meta-Review · Area_Chair_1E5M · 2026-01-11

**Summary:**

Reviewer 57Lz (score 4)
wanted the paper to include a discussion on FreeSurfer (that it is used in addition to the MR images). They also had a technical question on a two-stage pipeline which first performs volumetric deformation and then surface registration. They identified a limitation of the approach that it would typically need to rely on T1w images because the gray-white matter contrast there is strongest.

Reviewer 7rxY (score 2)
pointed to an existing paper that uses a similar approach.

Reviewer rb9L (score 4)
had comments that suggested modification of the narrative (showing evidence that volumetric registration fails for the cortex, optimization-based methods are slow). They also wanted to see more baselines. The main concern is that the problem formulation relies on using a volumetric grid with an objective that is computed on the cortical surface (this is a zero/near-zero measure part of the grid). So roughly, the premise of the paper regarding the inadequacy of volumetric registration is also evident in the present approach....it does not fix the issue at hand. The reviewer also pointed to a comments about related work that need to be fixed.

Reviewer L2kZ (score 8)
made comments about the method requiring cortical meshes, limited scope of the method (T1w), and wanted more ablations and clarifications.

This paper received detailed reviews and there was an elaborate response provided by the authors. Overall the sentiment of all reviewers (except 7rxY, who provided a very short review that lacked sufficient detail) are positive. I recommend that this paper be accepted.

**Reviewer Concerns:**

Reviewer 57Lz
The rebuttal addressed these comments adequately.

Reviewer 7rxY
The authors provided an elaborate response that included multiple existing papers of this kind. Their main argument is that the present manuscript is a modern instantiation of these ideas and that it works well because volumetric and surface registration are performed together.

Reviewer rb9L
The authors provided some new experiments (volumetric registration works poorly with cortical labelmaps, other baselines like FireAnts, VoxelMorph). Regarding the zero measure issue, the authors clarified that their implemented uses a trilinear interpolation inside the volumetric grid and this is why cortical residuals propagate to 8 different vertices in the grid to alleviate the issue of gradients being near zero. While this is not a complete solution or validated beyond the experiments in the paper, this is a neat implementation trick. The authors have addressed the other issues about the incorrect narrative.

Reviewer L2kZ
The authors have provided an elaborate response. All questions that were asked were adequately addressed, e.g., an extra figure, improving the math, details of the implementation, etc. Some of the technical comments were similar to the discourse with Reviewer rb9L.

**Reviewer Scores:**

Reviewer 57Lz
Post-rebuttal, I suspect the updated score would be more positive, e.g., 6.

Reviewer 7rxY
The review was very short and the score was exceedingly negative. I have discounted this review in my recommendation.

Reviewer rb9L
The updated score would be more positive, e.g., 6.

Reviewer L2kZ
The updated score would remain the same, 8.

---

### Decision · Program_Chairs · 2026-01-26

Accept (Poster)